# Understanding and Adopting Rational Behavior by Bellman Score Estimation

**Kuno Kim**
Department of Computer Science
Stanford University
Stanford, CA 94305
`khkim@cs.stanford.edu`

**Stefano Ermon**
Department of Computer Science
Stanford University
Stanford, CA 94305
`ermon@cs.stanford.edu`

## Abstract

We are interested in solving a class of problems that seek to understand and adopt rational behavior from demonstrations. We may broadly classify these problems into four categories of reward identification, counterfactual analysis, behavior imitation, and behavior transfer. In this work, we make a key observation that knowing how changes in the underlying rewards affect the optimal behavior allows one to solve a variety of aforementioned problems. To a local approximation, this quantity is precisely captured by what we term the *Bellman score*, i.e the gradient of the log probabilities of the optimal policy with respect to the reward. We introduce the Bellman score operator which provably converges to the gradient of the infinite-horizon optimal $Q$-values with respect to the reward which can then be used to directly estimate the score. Guided by our theory, we derive a practical *score-learning* algorithm which can be used for score estimation in high-dimensional state-actions spaces. We show that score-learning can be used to reliably identify rewards, perform counterfactual predictions, achieve state-of-the-art behavior imitation, and transfer policies across environments.

## 1 Introduction

A hallmark of intelligence is the ability to achieve goals with rational behavior. For sequential decision making problems, rational behavior is often formalized as a policy that is optimal with respect to a Markov Decision Process (MDP). In other words, intelligent agents are postulated to learn rational behavior for reaching goals by maximizing the reward of an underlying MDP (Doya, 2007; Neftci & Averbeck, 2019; Niv, 2009). The reward covers information about the goal while the remainder of the MDP characterizes the interplay between the agent's decisions and the environment.

In this work, we are interested in solving a class of problems that seek to understand and adopt rational (i.e optimal) behavior from demonstrations of sequential decisions. We may broadly classify them into categories of reward identification, counterfactual analysis, behavior imitation, and behavior transfer. (see Appendix D for detailed definitions) These four problems of understanding and adopting rationality arise across a wide spectrum of science fields. For example, in econometrics, the field of Dynamic Discrete Choice (DDC) seeks algorithms for fitting reward functions to human decision making behavior observed in labor (Keane & Wolpin, 1994) or financial markets (Arcidiacono & Miller, 2011). The identified utilities are leveraged to garner deeper insight into people's decision strategies (Arcidiacono & Miller, 2020), make counterfactual predictions about how their choices will change in response to market interventions that alter the reward function (Keane & Wolpin, 1997), and train machine learning models that can make rational decisions like humans (Kalouptsidi et al., 2015). In animal psychology and neuroscience, practitioners model the decision strategies of animals by fitting reward models to their observed behavior. The fitted reward is analyzed and then used to train AI models that simulate animal movements (Yamaguchi et al., 2018; Schafer et al., 2022) in order to gain a better understanding of ecologically and evolutionarily significant phenomena such as habitat selection and migration (Hirakawa et al., 2018). In robot learning, Inverse Reinforcement Learning (IRL) is used to infer reward functions from teleoperated robots and the learned rewards are fed through an RL algorithm to teach a robot how to perform the same task without human controls (Fu et al., 2018; Finn et al., 2016b; Chan & van der Schaar, 2021). As rewards for a task are often invariant to perturbations in the environment, (e.g walking can be described by the same reward function which encourages forward velocity and stability regardless of whether the agent is walking

on ice or grass) the same task can be learned in various environmental conditions by optimizing the same inferred reward (Fu et al., 2018; Zhang et al., 2018).

We make a key observation that knowing how changes in the underlying rewards affect the optimal behavior (policy) allows one to solve a variety of aforementioned problems. To a local approximation, this quantity is precisely captured by what we term the *Bellman score*, i.e *gradient of log probabilities of the optimal policy with respect to the reward*. Prior works which study related quantities to the Bellman score are scarce and relied on having full-knowledge of the environment dynamics or small discrete state-action spaces (Neu & Szepesvári, 2012; Vroman, 2014; Li et al., 2017). We introduce the Bellman score operator that provably converges to the gradient of the infinite-horizon optimal $Q$-values with respect to the reward. This $Q$-gradient can then be used to estimate the score. We further show that the $Q$-gradient is equivalent to the conditional state-action visitation counts under the optimal policy. With these results, we derive the gradient of the Maximum Entropy IRL (Ziebart et al., 2008; Finn et al., 2016b; Fu et al., 2018) objective in the general setting with stochastic dynamics case and non-linear reward models. Guided by theory, we propose a powerful score-learning algorithm that can be used for model-free score estimation in continuous high-dimensional state-action spaces and an effective IRL algorithm, named Gradient Actor-Critic (GAC). Our experiments demonstrate that score-learning can be used to reliably identify rewards, make counterfactual predictions, imitate behaviors, and transfer policies across environments.

## 2 PRELIMINARIES

A Markov Decision Process (MDP) $\mathcal{M} \in \Omega$, where $\Omega$ is the set of all MDPs, is a tuple $\mathcal{M} = (\mathcal{X}, \mathcal{A}, P, P_0, r_\theta, \gamma)$ where $\mathcal{X}$ is the discrete [1] state space, $\mathcal{A}$ is the discrete action (decision) space, $P \in \mathbb{R}^{|\mathcal{X} \times \mathcal{A}| \times |\mathcal{X}|}$ is the transition probability matrix, $P_0 \in \mathbb{R}^{|\mathcal{X}|}$ is the initial state distribution, $r_\theta \in \mathbb{R}^{|\mathcal{X} \times \mathcal{A}|}$ is the (stationary) parametric reward with parameters $\theta \in \Theta$, and $\gamma \in [0, 1]$ is the discount factor. $\Theta$ is the parameter space, typically set to a finite-dimensional real vector space $\mathbb{R}^{\dim(\theta)}$. We will use $T \geq 0$ to denote the time horizon for finite-horizon problems. A domain $d$ is an MDP without the reward $\mathcal{M} \setminus r$. Moving forward we will alternate between vector and function notation, e.g $P(x'|x, a), r_\theta(x, a)$ denote the value of vector $r_\theta$ at the dimension for the state-action $(x, a)$ and the value of matrix $P$ at the location for $((x, a), x')$. Furthermore, one-dimensional vectors are treated as row vectors, e.g $\mathbb{R}^{|\mathcal{X} \times \mathcal{A}|} = \mathbb{R}^{1 \times |\mathcal{X} \times \mathcal{A}|}$. A (stationary) policy is a vector $\pi \in \mathbb{R}^{|\mathcal{X} \times \mathcal{A}|}$ that represents distributions over actions, i.e $\sum_{a \in \mathcal{A}} \pi(a|x) = 1$ for all $x$. A non-stationary policy for time horizon $T$ is a sequence of policies $\tilde{\pi} = (\pi_t)_{t=0}^T \in (\mathbb{R}^{|\mathcal{X} \times \mathcal{A}|})^{T+1}$ where $\pi_t$ is the policy used when there are $t$ environment steps remaining [2] and $\tilde{\pi}_{:k} = (\pi_t)_{t=0}^k$ for $k \leq T$ denotes a subsequence. When there is no confusion we will use $\pi$ to denote to both stationary and non-stationary policies. Next, $P_\pi \in \mathbb{R}^{|\mathcal{X} \times \mathcal{A}| \times |\mathcal{X} \times \mathcal{A}|}$ denotes the transition matrix of the Markov chain on state-action pairs induced by policy $\pi$, i.e $P_\pi(x', a'|x, a) = P(x'|x, a)\pi(a'|x')$. Furthermore, $P_\pi^n$ denotes powers of the transition matrix with $P_\pi^0 = I$ where $I \in \mathbb{R}^{|\mathcal{X} \times \mathcal{A}| \times |\mathcal{X} \times \mathcal{A}|}$ is the identity. For non-stationary policies, we define $P_{\tilde{\pi}}^n = P_{\pi_{T-1}} P_{\pi_{T-2}} ... P_{\pi_{T-n}}$ for $n \geq 1$ and $P_{\tilde{\pi}}^0 = I$.

Let $\mathbb{1}_{x,a} \in \mathbb{R}^{|\mathcal{X} \times \mathcal{A}|}$ denote the indicator vector which has value 1 at the dimension corresponding to $(x, a)$ and 0 elsewhere. The conditional marginal distribution for both stationary and non-stationary policy $\pi$ after $n$ environment steps is $p_{\pi,n}(\cdot|x, a) = \mathbb{1}_{x,a} P_\pi^n \in \mathbb{R}^{|\mathcal{X} \times \mathcal{A}|}$. The (unnormalized) $t$-step[3] conditional occupancy measure of $\pi$ is the discounted sum of conditional marginals:

$$\rho_{\pi,t}(\cdot|x, a) = \sum_{n=0}^t \gamma^n p_{\pi,n}(\cdot|x, a) = \sum_{n=0}^t \gamma^n \mathbb{1}_{x,a} P_\pi^n . \tag{1}$$

Intuitively, $\rho_{\pi,t}(x', a'|x, a)$ quantifies the visitation frequency for $(x', a')$ when an agent starts from $(x, a)$ and runs $\pi$ for $t$ environment steps, with more weight on earlier visits. The infinite horizon conditional occupancy exists for $\gamma < 1$ and we will denote it by simply omitting the subscript $t$, i.e $\rho_\pi(\cdot|x, a) = \lim_{t \to \infty} \rho_{\pi,t}(\cdot|x, a)$. The (unconditional) occupancy measure can be recovered by $\rho_{\pi,t}(x', a') = \sum_{x,a} P_0(x)\pi_T(a|x)\rho_{\pi,t}(x', a'|x, a)$. The $t$-step $Q$-values $Q_{\pi,t} \in \mathbb{R}^{|\mathcal{X} \times \mathcal{A}|}$ for both stationary and non-stationary policy $\pi$ are defined as $Q_{\pi,t}(x, a) = \mathbb{E}_{x',a' \sim \rho_{\pi,t}(\cdot|x,a)}[r_\theta(x', a')] = \rho_{\pi,t}(\cdot|x, a) \cdot r$, i.e the conditional expectation of discounted reward sums when there are $t$ environment

---

[1] We use discrete spaces for notational brevity later on, but our results can be extended to continuous spaces

[2] Backwards indexing is standard in dynamic programming RL algorithms (Bertsekas & Tsitsiklis, 1995)

[3] *Step* refers to the number of environment transitions and not the number of decisions made

steps remaining. The theory in this section will be derived in the Maximum Entropy RL (MaxEntRL) setting (Haarnoja et al., 2017; Yarats et al., 2021). Let us define the $t$-step discounted conditional entropy of a policy as $\mathcal{H}_{\pi,t}(x,a) = -\sum_{n=1}^{t} \gamma^n p_{\pi,n}(\cdot|x,a) \cdot \log \pi_{T-n}$. In MaxEntRL, the soft $Q$-values are defined as $Q_{\pi,t}^{\text{soft}}(x,a) = Q_{\pi,t}(x,a) + \mathcal{H}_{\pi,t}(x,a)$, i.e the conditional expectation of discounted reward and policy entropy sums.

For reward $r_\theta$, let $Q_{\theta,t}^{\text{soft}} \in \mathbb{R}^{|\mathcal{X}\times\mathcal{A}|}$ denote the $t$-step optimal $Q$-values defined iteratively by the soft Bellman optimality operator $\mathcal{T}_{\mathcal{H}} : \mathbb{R}^{|\mathcal{X}\times\mathcal{A}|} \to \mathbb{R}^{|\mathcal{X}\times\mathcal{A}|}$ as

$$Q_{\theta,t}^{\text{soft}}(x,a) = (\mathcal{T}_{\mathcal{H}}Q_{\theta,t-1}^{\text{soft}})(x,a) = r_\theta(x,a) + \gamma\mathbb{E}_{x'\sim P(\cdot|x,a)}[\log\sum_{a'\in\mathcal{A}}\exp Q_{\theta,t-1}^{\text{soft}}(x',a')]. \quad (2)$$

for all $t > 0$ and $Q_{r,0}^{\text{soft}} = r$. Hence the optimal $Q$-values for a decision problem with horizon $t$ can be obtained by sequentially computing the optimal $Q$-values for $1, ..., t-1$ step decision problems using dynamic programming (Bellman, 1957), i.e value iteration. The operator $\mathcal{T}_{\mathcal{H}}$ is a contraction in the max-norm (Bertsekas & Tsitsiklis, 1995) whose unique fixed point is the infinite-horizon optimal $Q$-values $Q_{\theta,\infty}^{\text{soft}}$ satisfying $Q_{\theta,\infty}^{\text{soft}} = \mathcal{T}_{\mathcal{H}}Q_{\theta,\infty}^{\text{soft}}$. For finite horizon problems, the (non-stationary) optimal policy $\pi_\theta^{\text{soft}} = (\pi_{\theta,t}^{\text{soft}})_{t=0}^{T}$ is derived by $\pi_{\theta,t}^{\text{soft}}(a|x) = \text{softmax}(Q_{\theta,t}^{\text{soft}})(x,a) = e^{Q_{\theta,t}^{\text{soft}}(x,a)}/\sum_a e^{Q_{\theta,t}^{\text{soft}}(x,a)}$ and, similarly, the infinite-horizon (stationary) optimal policy is $\pi_{\theta,\infty}^{\text{soft}} = \text{softmax}(Q_{\theta,\infty}^{\text{soft}})$. We may now define the key quantity of interest, the *Bellman score*.

**Definition 1** *The finite-horizon **Bellman score** $s_\theta^{\text{soft}} \in (\mathbb{R}^{|\mathcal{X}\times\mathcal{A}|\times\dim(\theta)})^{T+1}$ is the gradient of the log probabilities of the (non-stationary) optimal policy with respect to the reward,*

$$s_\theta^{\text{soft}} = (s_{\theta,t}^{\text{soft}})_{t=0}^{T} = (\nabla_\theta\log\pi_{\theta,t}^{\text{soft}})_{t=0}^{T}.$$

*Similarly the infinite-horizon Bellman score $s_{\theta,\infty}^{\text{soft}} \in \mathbb{R}^{|\mathcal{X}\times\mathcal{A}|\times\dim(\theta)}$ is*

$$s_{\theta,\infty}^{\text{soft}} = \nabla_\theta\log\pi_{\theta,\infty}^{\text{soft}}.$$

In words, the finite-horizon score $s_{\theta,t}^{\text{soft}}$ is the Jacobian of the log optimal policy vector with respect to the reward parameters, i.e $s_{\theta,t}^{\text{soft}}(x,a) = \nabla_\theta\log\pi_{\theta,t}^{\text{soft}}(a|x)$ is the direction in which the reward parameters $\theta$ should perturbed in order to increase the log probability of action $a$ at state $x$ under the optimal policy for time $t$. To a local linear approximation around $\theta$, the score conveys how the optimal behavior changes as a function of the underlying reward parameters. Next, we will prove useful properties of the score as well as derive a dynamic programming algorithm to compute it.

## 3 SCORE ITERATION

Since $\pi_{\theta,t}^{\text{soft}}(a|x) = \text{softmax}(Q_{\theta,t}^{\text{soft}})(x,a)$, the Bellman score can be written as a difference of the gradients of optimal $Q$-values with respect to the reward parameters:

$$s_{\theta,t}^{\text{soft}}(x,a) = \nabla_\theta\log\pi_{\theta,t}^{\text{soft}}(a|x) = \nabla_\theta Q_{\theta,t}^{\text{soft}}(x,a) - \mathbb{E}_{a'\sim\pi_{\theta,t}^{\text{soft}}(\cdot|x)}[\nabla_\theta Q_{\theta,t}^{\text{soft}}(x,a')]. \quad (3)$$

We will refer to $\nabla_\theta Q_{\theta,t}^{\text{soft}}$ as the $Q$-gradient or value gradient. Eq. 3 shows that an unbiased sample estimate of the score can be obtained from the $Q$-gradient and the optimal policy. Thus, we will derive a dynamic programming algorithm termed *score iteration* which efficiently computes the $Q$-gradients. Similar to how value iteration proceeds by repeated application of the Bellman optimality operator (Eq. 2), score iteration will rely on the Bellman score operator which we define now.

**Definition 2** *The **Bellman score operator** $\mathcal{G}_{\pi,\theta} : \mathbb{R}^{|\mathcal{X}\times\mathcal{A}|\times\dim(\theta)} \to \mathbb{R}^{|\mathcal{X}\times\mathcal{A}|\times\dim(\theta)}$ for a policy $\pi$ and reward $r_\theta$ is defined on an input $J \in \mathbb{R}^{|\mathcal{X}\times\mathcal{A}|\times\dim(\theta)}$ as*

$$\mathcal{G}_{\pi,\theta}J = \nabla_\theta r_\theta + \gamma P_\pi J.$$

To gain intuition about the score operator, consider the tabular reward $r_\theta(x,a) = \theta(x,a)$ where $\theta \in \mathbb{R}^{|\mathcal{X}\times\mathcal{A}|}$. In this setting, the score operator simply computes the conditional occupancies for $\pi$ (Eq. 1) when applied to the identity matrix $I$. For example, the values of $\mathcal{G}_{\pi,\theta}I = \nabla_\theta r_\theta + \gamma P_\pi I = I + \gamma P_\pi I$ at the row for state-action $(x,a)$ is $\mathbb{1}_{x,a} + \gamma P_\pi(\cdot|x,a)$ which corresponds to the one-step conditional occupancy $\rho_{\pi,1}(\cdot|x,a)$. While the score operator has a pleasantly simple form, Theorem 1 will show that the $Q$-gradients can be computed by its repeated application and, as a result, that the $Q$-gradient is the conditional occupancy measure of the optimal policy for the underlying reward.

---

**Algorithm 1** Score Iteration: Computes the infinite-horizon Bellman score via dynamic programming

---

$\mathcal{M} = (\mathcal{X}, \mathcal{A}, P, P_0, r_\theta, \gamma)$: Markov Decision Process
$g$: randomly initialized $Q$-gradient vector
**procedure** SCOREITERATION($\mathcal{M}$)
    For $\mathcal{M}$, learn the optimal policy $\pi_{\theta,\infty}^{\text{soft}}$
    **while** $g$ is not converged **do**
        **for** $x \in \mathcal{X}, a \in \mathcal{A}$ **do**
            Update $Q$-gradient: $g(x,a) \leftarrow \nabla_\theta r_\theta(x,a) + \gamma \sum_{x',a' \in \mathcal{X} \times \mathcal{A}} P_{\pi_{\theta,\infty}^{\text{soft}}}(x',a'|x,a)g(x',a')$
    **for** $x \in \mathcal{X}, a \in \mathcal{A}$ **do**
        Compute Bellman score: $s(x,a) \leftarrow g(x,a) - \sum_{a' \in \mathcal{A}} \pi_{\theta,\infty}^{\text{soft}}(a'|x)g(x,a')$
    **return** $\pi_{\theta,\infty}^{\text{soft}}, g, s$

---

**Theorem 1** *For all $t = 1, ..., T$ and any matrix $J \in \mathbb{R}^{|\mathcal{X} \times \mathcal{A}| \times \dim(\theta)}$*

$$\nabla_\theta Q_{\theta,t}^{\text{soft}} = \mathcal{G}_{\pi_{\theta,t-1}^{\text{soft}},\theta} ... \mathcal{G}_{\pi_{\theta,0}^{\text{soft}},\theta}(\nabla_\theta Q_{\theta,0}^{\text{soft}})$$

$$\nabla_\theta Q_{\theta,\infty}^{\text{soft}} = \mathcal{G}_{\pi_{\theta,\infty}^{\text{soft}},\theta}(\nabla_\theta Q_{\theta,\infty}^{\text{soft}}) = \lim_{k \to \infty} \mathcal{G}_{\pi_{\theta,\infty}^{\text{soft}},\theta}^k J$$

*where, $\nabla_\theta Q_{\theta,0}^{\text{soft}} = \nabla_\theta r_\theta$. Furthermore, the $Q$-gradient satisfies:*

$$\nabla_\theta Q_{\theta,t}^{\text{soft}}(x,a) = \mathbb{E}_{x',a' \sim \rho_{\pi_{\theta,:t}^{\text{soft}},t}(\cdot|x,a)}[\nabla_\theta r_\theta(x',a')]$$

$$\nabla_\theta Q_{\theta,\infty}^{\text{soft}}(x,a) = \mathbb{E}_{x',a' \sim \rho_{\pi_{\theta,\infty}^{\text{soft}}}(\cdot|x,a)}[\nabla_\theta r_\theta(x',a')]$$

Theorem 1 shows that both finite and infinite horizon $Q$-gradients can be computed by repeatedly applying the score operator of the current reward $r_\theta$ and its optimal policy $\pi_\theta^{\text{soft}}, \pi_{\theta,\infty}^{\text{soft}}$. Similar to how the Bellman optimality operator (Eq. 2) converges any starting vector in $\mathbb{R}^{|\mathcal{X} \times \mathcal{A}|}$ to the infinite-horizon optimal $Q$-values (Bertsekas & Tsitsiklis, 1995), the Bellman score operator converges any starting matrix in $\mathbb{R}^{|\mathcal{X} \times \mathcal{A}| \times \dim(\theta)}$ to the infinite-horizon $Q$-gradient.

Furthermore, we see that the Q-gradient is in fact the expected reward gradient under the conditional occupancy measure. To gain more intuition, once again consider the tabular reward $r_\theta(x,a) = \theta(x,a)$. Then, $\nabla_\theta Q_{\theta,t}^{\text{soft}}(x,a) = \mathbb{E}_{x',a' \sim \rho_{\pi_{\theta,:t}^{\text{soft}},t}(\cdot|x,a)}[\mathbb{1}_{x',a'}] = \rho_{\pi_{\theta,:t}^{\text{soft}},t}$. We see that the $Q$-gradient is in fact equivalent to the conditional occupancy of the optimal policy, which is consistent with our previous analysis of the score operator. Thus, in order to increase the optimal Q-values at $(x,a)$, the rewards of state-actions $(x',a')$ should be increased proportional to their discounted visitation frequencies when starting from $(x,a)$ and following the optimal policy. As an alternative to the operator perspective, one may consider a Linear Programming (LP) interpretation. The optimal $Q$-values are the solution to an LP optimization problem over the set of conditional occupancies $D$ that are feasible under the Markovian environment dynamics, i.e $Q_{\theta,t}^{\text{soft}}(x,a) = \max_{\rho \in D} r_\theta \cdot \rho$. Loosely speaking, invoking the supremum rule (Boyd et al., 2003) allows one to show that the gradient of the LP solution, i.e $\nabla_{r_\theta} \max_{\rho \in D} r_\theta \cdot \rho$, is in fact the conditional occupancy that solves the LP.

*The main practical contribution of Theorem 1 is that the operator convergence results enable designing a powerful dynamic programming method for score computation as shown in Algorithm 1.* (Finite-horizon version in Appendix D) The algorithm computes the optimal policy $\pi_{\theta,\infty}^{\text{soft}}$, then proceeds by repeatedly applying the score operator $\mathcal{G}_{\pi_{\theta,\infty}^{\text{soft}},\theta}$. While highly effective in countable state-action spaces, score iteration has limited applicability to continuous state-action spaces and requires full knowledge of the environment dynamics. Similar to how $Q$-learning is a model-free instantiation of the value iteration algorithm, we will proceed to propose score-learning which is a model-free instantiation of the score iteration algorithm.

## 4 SCALING UP SCORE ESTIMATION

In order to scale score estimation to the model-free settings with (potentially) high-dimensional continuous state-action spaces, we start by parameterizing the policy, $Q$-gradient network, and score network as neural networks $\pi_\phi : \mathcal{X} \times \mathcal{A} \to \mathbb{R}$, $g_\psi : \mathcal{X} \times \mathcal{A} \to \mathbb{R}^{\dim(\theta)}$, and $s_\omega : \mathcal{X} \times \mathcal{A} \to \mathbb{R}^{\dim(\theta)}$. Just as $Q$-learning (Silver et al., 2016) approximates an application of the Bellman optimality operator by learning a $Q$-network that minimizes a boot-strapped regression loss, score-learning approximates

---

**Algorithm 2** Score-learning: model-free instantiation of SCOREITERATION

---

$(\theta, \phi, \psi, \omega)$: Weights for reward, policy, $Q$-gradient, and score network
$\mathcal{B}$: Buffer of transitions (can be either online or offline data)
$N, N_\psi$: Total number of algorithm iterations, number of $Q$-gradient update steps per policy update
$\eta_\psi, \eta_\omega$: $Q$-gradient learning rate, score-learning rate
$\alpha_g$: Target $Q$-gradient mixing rate
RL: One policy update step in an RL algorithm, e.g one policy gradient step in SAC (Haarnoja et al., 2018).
**procedure** SCORELEARNING($\theta, \phi, \psi, \omega, \mathcal{B}, N$)
    **for** $i \in \{1, \ldots, N\}$ **do**
        *# Update policy $\pi_\phi$*
        **if** $\phi$ is not None **then**
            Update policy to maximize reward $r_\theta$: $\phi \leftarrow \text{RL}(\theta, \phi, \mathcal{B})$
        *# Update $Q$-gradient $g_\psi$*
        **for** $j \in \{1, \ldots, N_\psi\}$ **do**
            Sample batch: $(x, a, x') \sim \mathcal{B}$
            **if** $\psi$ is not None **then**
                Sample next action from current policy: $a' \sim \pi_\phi(\cdot|x')$
            **else**
                Sample next action from buffer $a' \sim \mathcal{B}$
            Update $Q$-gradient (Eq. 4): $\psi \leftarrow \psi - \eta_\psi \nabla_\psi (\nabla_\theta r_\theta(x, a) + \gamma \bar{g}_\psi(x', a') - g_\psi(x, a))^2$
            Update Target $Q$-gradient $\bar{g}_\psi$ with soft mixing rate $\alpha_\psi$ (Haarnoja et al., 2017)
        *# Update Bellman score $s_\omega$*
        Sample batch and contrastive action: $(x, a) \sim \mathcal{B}, a' \sim \pi_\phi(\cdot|x)$
        Update score network (Eq. 5): $\omega \leftarrow \omega - \eta_s \nabla_\omega (g_\psi(x, a) - g_\psi(x, a') - s_\omega(x, a))^2$
    **return** $\phi, \psi, \omega$

---

an application of the score operator by learning a $Q$-gradient network that minimizes the boot-strapped regression loss in Eq. 4. The score network then minimizes the regression loss in Eq. 5 which directly follows by replacing the $Q$-gradient with our estimate $g_\psi$ in Eq. 3.

$$L_\psi = (\nabla_\theta r_\theta(x, a) + \gamma \mathbb{E}_{x', a' \sim P_{\pi_\phi}(\cdot|x, a)}[\bar{g}_\psi(x', a')] - g_\psi(x, a))^2 \tag{4}$$

$$L_\omega = (g_\psi(x, a) - \mathbb{E}_{a' \sim \pi_\phi(\cdot|x)}[g_\psi(x, a')] - s_\omega(x, a))^2 \tag{5}$$

Note that $\bar{g}_\psi$ is a target $Q$-gradient network used for stabilizing optimization analogous to target $Q$-networks that are used in $Q$-learning (Haarnoja et al., 2017). Algorithm 2 shows the full execution flow of score-learning. Instead of sequentially learning each model until convergence, score-learning alternates between making small improvements to the policy (red), regressing to the $Q$-gradient with the current estimate of the optimal policy (blue), and updating the score (yellow) with the current estimate of the $Q$-gradient. This enables the algorithm to output approximate estimates of all three components when run for a few-steps, which will be useful for downstream algorithms that use approximate score-learning in the inner-loop. If the buffer $\mathcal{B}$ already contains samples from the optimal policy, simply not passing $\phi$ disables policy updates. Note that score-learning can operate with both online and offline data $\mathcal{B}$ depending on the choice of the RL algorithm used to update the policy. We now show how the score-learning algorithm can be used for a variety of downstream tasks.

## 4.1 MAXIMUM ENTROPY INVERSE REINFORCEMENT LEARNING WITH SCORE-LEARNING

In this section we will show how score-learning can be used for maximum entropy inverse reinforcement learning (MaxEntIRL). Let $\tau = (x_0, a_0, ..., x_T, a_T)$ denote a trajectory of state-actions, and $p_\theta(\tau) = \log P_0(x_0) \prod_{t=0}^{T} \pi_{\theta, t}^{\text{soft}}(a_t|x_t) \prod_{t=0}^{T-1} P(x_{t+1}|x_t, a_t)$ denote the trajectory distribution of the MaxEnt optimal policy $\pi_\theta^{\text{soft}}$ for reward $r_\theta$ and similarly let $p^*(\tau)$ be the trajectory distribution of the expert policy $\pi^* = (\pi_t^*)_{t=0}^{T}$ from which the demonstrations $\mathcal{D}$ are sampled. The goal in MaxEntIRL is to find reward parameters $\theta$ that maximize the log-likelihood of the expert trajectories:

$$\max_\theta \mathbb{E}_{\tau \sim p^*}[\log p_\theta(\tau)]. \tag{6}$$

The gradient of the objective in Eq. 6 is highly useful as it enables direct gradient ascent to solve the maximization. Previously, the MaxEntIRL gradient has been derived in either the limited setting of deterministic dynamics (Ziebart et al., 2008) or in the general stochastic dynamics setting for the special case of linear reward models (Ziebart et al., 2010). Here, we leverage Theorem 1 to derive the general MaxEntIRL gradient in the stochastic dynamics setting with non-linear rewards.

---

**Algorithm 3** Gradient Actor-Critic (GAC ): Inverse RL via $Q$-gradient Estimation

---

$(\theta, \phi, \psi_L, \psi_E)$: Weights for reward, policy, learner $Q$-gradient, and expert $Q$-gradient network
$\mathcal{D}$: Demonstrations of expert behavior
$M$: Total number GAC iterations
$N, N_\theta$: Number of score iteration steps per GAC iteration, Reward update interval
$\eta_r$: Reward learning rate
**procedure** GAC$(\theta, \phi, \bar{\phi}, \psi, \mathcal{D})$
    **for** $i \in \{1, \dots, M\}$ **do**
        Take step in environment with $\pi_\phi$, add transition $(x, a, s')$ to buffer $\mathcal{B}$
        Update learner $Q$-gradient and learner policy:
            $\phi, \psi_L \leftarrow$ SCORELEARNING$(\theta, \phi, \psi_L, \mathcal{B}, N)$
        Update expert $Q$-gradient:
            $\psi_E \leftarrow$ SCORELEARNING$(\theta, \phi$=None$, \psi_E, \mathcal{D}, N)$
        **if** $i \mod N_\theta == 0$ **then**
            Sample expert batch of initial state-actions $(x_0, a_0) \sim \mathcal{D}_{\mathrm{demo}}$
            Sample learner batch of initial state-actions $\tilde{x}_0 \sim \mathcal{B}, \tilde{a}_0 \sim \pi_\phi(\cdot|\tilde{x}_0)$
            Update reward with $Q$-gradient: $\theta \leftarrow \theta + \eta_r(g_\psi(x_0, a_0) - g_\psi(\tilde{x}_0, \tilde{a}_0))$
    **return** $\theta, \phi$

---

**Theorem 2** *The gradient of the MaxEntIRL objective of Eq. 6 is*

$$\nabla_\theta \mathbb{E}_{\tau \sim p^*}[\log p_\theta(\tau)] = \mathbb{E}_{x \sim P_0, a \sim \pi_T^*(\cdot|x), a' \sim \pi_{\theta,T}^{\mathrm{soft}}(\cdot|x)}[\nabla_\theta Q_{\pi^*,T}(x, a) - \nabla_\theta Q_{\theta,T}(x, a')] \quad (7)$$

$$= \mathbb{E}_{x,a \sim \rho_{\pi^*,T}}[\nabla_\theta r_\theta(x, a)] - \mathbb{E}_{x',a' \sim \rho_{\pi_\theta^{\mathrm{soft}},T}}[\nabla_\theta r_\theta(x', a')]. \quad (8)$$

Theorem 2 shows that the reward should be updated to increase the $Q$-values of the expert and decrease the optimal $Q$-values for the learner (Eq. 7) which is equivalent to increasing the reward on state-actions sampled from the expert and decreasing the reward of state-actions sampled from the learner (Eq. 8). With deterministic dynamics, Eq. 8 simply reduces to the known contrastive divergence gradient (Finn et al., 2016b) for energy-based models. When $r_\theta$ is linear, Eq. 8 reduces to the expected feature gap derived in Ziebart et al. (2010). (see Appendix A for extended discussion)

Many prior works in MaxEntIRL (Finn et al., 2016b; Fu et al., 2018) take Monte Carlo samples to estimate Eq. 8 and perform stochastic gradient updates on $\theta$. The key challenges with this approach are, first, the difficulty of reusing old samples $(x, a) \sim \pi_{\theta_{\mathrm{old}}}^{\mathrm{soft}}$ from policies for older rewards $\theta_{\mathrm{old}}$ and, second, the high variance of the Monte Carlo estimator. *The main practical significance of Theorem 2 comes from Eq. 7 which reveals that we may instead learn $Q$-gradient networks with score-learning to approximate the expectations in Eq. 8.* At the cost of introducing bias to the gradient estimates, this approach resolves the two aforementioned challenges; score-learning works with offline data, which enables old sample reuse, and the $Q$-gradient network outputs the mean reward gradient, which reduces variance. We name our algorithm Gradient Actor-Critic (GAC ) (see Algorithm 3) which uses score-learning to estimate the expert $Q$-gradient $\nabla_\theta Q_{\pi^*,T}(x, a)$ and learner $Q$-gradient $\nabla_\theta Q_{\theta,T}(x, a)$ with parametric functions $g_{\psi_E}$ and $g_{\psi_L}$. To reduce computational cost, GAC alternates between approximate $Q$-gradient estimation, i.e running score-learning for $N$ steps, and using the rough gradients for reward updates. Our method is analogous to actor-critic methods (Haarnoja et al., 2017) that learn a $Q$-network to estimate mean policy returns which are then used for policy gradient updates. These methods similarly introduce bias into the policy gradients in exchange for the ability to reuse old policy data and perform lower variance gradient updates. Just as how actor-critic typically outperforms Monte-Carlo policy gradient methods, our experiments will demonstrate a similar usefulness in trading off bias for lower variance.

## 4.2 COUNTERFACTUAL PREDICTIONS WITH SCORE-LEARNING

Here we describe how score-learning can be used for counterfactual predictions. The goal is to predict how the optimal policy changes when the original reward $r_\theta$ is perturbed to a counterfactual reward $r_{\theta'}$. With access to an efficient environment simulator, one natural solution is to simply resolve the RL problem $r_{\theta'}$. However, this is difficult in many domains where solving the RL problem is expensive, particularly when we wish to make many counterfacutal predictions for multiple reward alterations. Instead, we may use the score $s_{\theta,\infty}^{\mathrm{soft}}$ for the original reward $r_\theta$ to estimate of the optimal policy at the new reward $r_{\theta'}$ by simply using the first order taylor approximation of the log optimal policy:

$$\log \pi_{\theta',\infty}^{\mathrm{soft}}(x, a) \approx \log \pi_{\theta,\infty}^{\mathrm{soft}}(x, a) + s_{\theta,\infty}^{\mathrm{soft}}(x, a) \cdot (\theta' - \theta). \quad (9)$$

Eq. 9 shows that $\log \pi_{\theta',\infty}^{\mathrm{soft}}(a|x)$ increases proportional to how well the direction of change in reward $(\theta' - \theta)$ aligns with the score $s_\theta(x, a)$. We use Algorithm 2 to estimate $s_\theta$ with $g_\psi$ and apply Eq. 9.

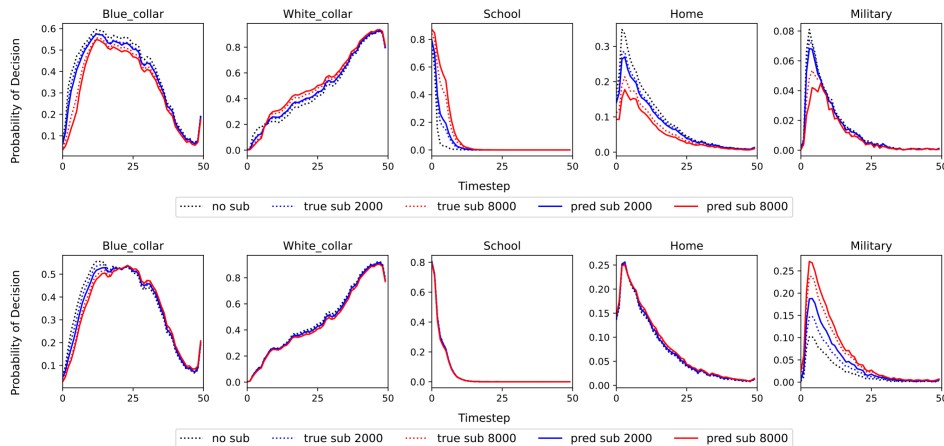

Figure 1: **Counterfactual Prediction Performance** for two scenarios: education subsidy (top row) and military enrollment incentives (bottom row). "no sub" corresponds to the initial reward from which the scores are estimated, "true sub" shows the probability of actions under the MaxEnt optimal policy obtained by resolving the RL model for perturbed rewards, and "pred sub" shows the score-estimated optimal policy probabilities. $2000, 8000$ denote the magnitude of the perturbation to the reward parameters which corresponds to the amount of subsidy or incentives.

## 5 EXPERIMENTS

### 5.1 REWARD IDENTIFICATION AND COUNTERFACTUAL PREDICTIONS

**Environment**: We experiment with the Keane and Wolpin (KW) environment (Keane & Wolpin, 1994; 1997) which is a widely used benchmark in the DDC literature. The KW environment simulates a life-time of occupational decisions where at each time-step an agent selects among five actions: white collar work, blue collar work, military, school, and staying home. The state is a 25 dimensional vector consisting of various features about the agent such as the time spent in each occupation, and their current job. The dynamics captures how their current job decision alters the state. Agents are required to be forward looking and maximize the rewards summed over a life-time of occupational choices. The reward parameter $\theta$ is a 63 dimensional vector where each parameter dimension corresponds to either a cost or benefit, e.g the cost of attending college or the benefit of transitioning from a white collar job to the military. The specific structure of the reward is omitted for brevity, but full details about the environment can be found in (Keane & Wolpin, 1994; 1997).

**Results**: We solve the forward MaxEntRL problem in the KW environment and use the optimal policy to generate 100 trajectories of occupational choices for two different reward parameters: $\theta_{edu}$ and $\theta_{mil}$. The first reward simulates a scenario with lower education costs and the latter captures a scenario with high benefits for military enrollment (Keane & Wolpin, 1994; 1997).

Table 1: **Reward Identification on KW dataset.** Mean squared error between true reward and IRL estimated reward evaluated on the demonstrations.

|  | GCL | MSM | GAC |
|---|---|---|---|
| $\theta_{edu}$ | $12.1 \pm 5.3$ | $\mathbf{0.14 \pm 0.4}$ | $\mathbf{0.16 \pm 0.3}$ |
| $\theta_{mil}$ | $20.5 \pm 6.8$ | $0.14 \pm 0.3$ | $\mathbf{0.08 \pm 0.2}$ |

We test the reward identification performance by computing the mean squared error between the ground truth reward $r_{\theta_{edu}}, r_{\theta_{mil}}$ and the estimated rewards on $r_{\hat{\theta}_{edu}}, r_{\hat{\theta}_{mil}}$ evaluated on the demonstrations. As rewards are only identifiable up to additive constant shifts (Kim et al., 2021), we search this equivalence class for the parameters that minimize the mean squared error. Table 1 shows that GAC is able to identify the rewards to similar precision as the Method of Simulated Moments (MSM) baseline which is the state-of-the-art method for identification in the DDC literature. While GAC and GCL are optimizing the same objective, i.e Eq. 8, we posit that trading off bias for lower variance (as explained in section 4.1) accounts for GAC's superior performance.

Next we test the counterfactual prediction performance using the method in section 4.2. We simulate two counterfactual scenarios where the government provides a subsidy for college tuition and military participation. The scenarios are simulated by reducing the cost parameters in the reward for attending college and joining the military. Figure 1 shows how our score-estimated counterfactual predictions of policy changes compare to fully re-solving the RL problem for the perturbed rewards. We are able

Table 2: **State Control Performance** when provided with *one* demonstration.

|  | BC | DAC | AIRL | GCL | GAC (OURS) |
|---|---|---|---|---|---|
| BALL IN CUP CATCH | $39.2 \pm 15.2$ | $291.9 \pm 29.1$ | $124.3 \pm 25.0$ | $502.2 \pm 104.2$ | $\mathbf{949.9 \pm 42.4}$ |
| CARTPOLE SWING-UP | $106.0 \pm 50.1$ | $709.0 \pm 40.9$ | $591.4 \pm 134.2$ | $320.1 \pm 105.8$ | $\mathbf{970.1 \pm 26.1}$ |
| FINGER SPIN | $159.1 \pm 60.1$ | $203.5 \pm 91.9$ | $105.0 \pm 22.8$ | $304.1 \pm 50.6$ | $\mathbf{984.0 \pm 18.4}$ |
| FISH SWIM | $205.9 \pm 49.2$ | $150.5 \pm 10.8$ | $10.5 \pm 85.1$ | $203.0 \pm 89.4$ | $\mathbf{946.8 \pm 22.5}$ |
| HOPPER STAND | $10.6 \pm 50.2$ | $809.1 \pm 98.4$ | $592.2 \pm 101.2$ | $\mathbf{900.1 \pm 150.2}$ | $917.1 \pm 15.9$ |
| WALKER STAND | $143.5 \pm 22.8$ | $406.5 \pm 89.5$ | $739.6 \pm 63.5$ | $194.4 \pm 39.5$ | $\mathbf{990.0 \pm 49.0}$ |
| WALKER WALK | $40.1 \pm 39.1$ | $125.9 \pm 78.1$ | $495.1 \pm 77.2$ | $102.4 \pm 30.0$ | $\mathbf{943.1 \pm 56.7}$ |
| CHEETAH RUN | $230.1 \pm 99.8$ | $206.10 \pm 75.1$ | $102.4 \pm 29.5$ | $169.1 \pm 20.3$ | $\mathbf{807.0 \pm 17.5}$ |
| QUADRUPED RUN | $10.45 \pm 91.9$ | $316.3 \pm 33.6$ | $195.4 \pm 91.5$ | $33.5 \pm 98.2$ | $\mathbf{961.3 \pm 50.1}$ |

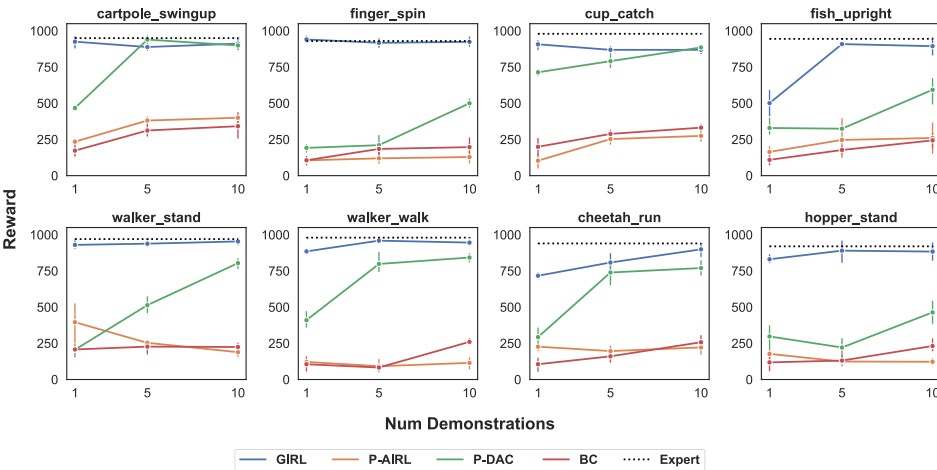

Figure 2: **Visual Control Performance** when a varying number of demonstrations.

to accurately estimate the increase in time spent in school and the military. Particularly, when the reward perturbations are smaller, the score-based counterfactual predictions are more accurate. For significantly larger perturbations, we would expect our method's accuracy to drop as the first order approximation is no longer valid.

## 5.2 BEHAVIOR IMITATION

**Environment and Baselines**: We experiment with high-dimensional robotics control environments from the DeepMind Control Suite (DMC) (Tassa et al., 2018), which is a widely used benchmark for RL and Imitation Learning (IL). For state control problems the agent has access to its proprioceptive states. For visual control the agent sees three consecutive observation images of size $84 \times 84 \times 3$ and uses action-repeat following the standard DMC protocol for visual RL (Yarats et al., 2022). We compare GAC against Behavioral Cloning (BC) (Pomerleau, 1991), Adversarial IRL (AIRL) (Fu et al., 2018), Guided Cost Learning (GCL) (Finn et al., 2016b), and Discriminator Actor-Critic (DAC) (Kostrikov et al., 2019; Cohen et al., 2021). We report the mean performance and standard deviation across ten random seeds where performance of each seed is measured as the mean reward of twenty trajectories sampled from the final policy. (see Appendix B for experiment details)

**Results**: For state control, we experiment in the extreme low-data regime where only one demonstration trajectory (consisting of 1000 state-actions) is provided. The experts for all tasks obtain rewards near 1000. Table 2 shows that GAC outperforms all baselines while achieving expert level performance. For visual control, Figure 2 shows that GAC outperforms baselines while achieving expert performance on many hard visual control tasks with just one demonstration. In particular, we found that the IRL baselines GCL and AIRL are incapable of performing well, which suggests that trading off bias for lower variance is beneficial in practice. While DAC achieves competitive performance on some tasks such as HOPPER STAND, it does not have the benefit of learning a reward. Some limitations of GAC include its longer compute time, potential optimization instabilities from the deadly triad (Van Hasselt et al., 2018), and double-sampling (Zhu & Ying, 2020). A more complete discussion of the limitations as well as hyper-parameter ablation studies are in Appendix C.

Table 3: **Reward Transfer** when provided with *ten* demonstrations in the original environment

|  | DIRECT | AIRL | GCL | GAC (OURS) |
|---|---|---|---|---|
| SLIPPERY WALKER | $302.5 \pm 29.5$ | $493.2 \pm 49.3$ | $201.9 \pm 89.6$ | $\mathbf{789.3 \pm 93.3}$ |
| STIFF WALKER | $205.2 \pm 98.3$ | $455.2 \pm 99.3$ | $301.3 \pm 89.4$ | $\mathbf{668.9 \pm 60.2}$ |
| SLIPPERY CHEETAH | $366.5 \pm 29.3$ | $591.9 \pm 65.7$ | $93.1 \pm 60.1$ | $\mathbf{803.9 \pm 98.9}$ |
| STIFF CHEETAH | $363.2 \pm 39.6$ | $430.3 \pm 68.2$ | $325.0 \pm 44.6$ | $\mathbf{650.3 \pm 92.3}$ |

## 5.3 BEHAVIOR TRANSFER

Next we show that that learned rewards from GAC robustly transfer behaviors to perturbed environments (Fu et al., 2018). We create variations of the state-based DMC environments by altering the surface and joint frictions to simulate realistic scenarios for robot learning where we may want to learn a single reward that can teach robots of different joint mechanics to mobilize in different walking surfaces. For example, the SLIPPERYWALKER environment has about $80\%$ less friction on the walking surface while STIFFCHEETAH has roughly $150\%$ higher friction on the robot joints. Rewards are first learned on the non-perturbed environments using 10 demonstrations, then re-optimized via RL on the perturbed environments. As a sanity check, we also include the DIRECT baseline which simply uses policy learned in the non-perturbed environment. Table 3 shows that GAC is able to outperform the baselines. However, GAC is not able to attain perfect expert level performance, which we believe is due to the perturbed environments requiring slightly different state visitation strategies.

## 6 RELATED WORKS

**Inverse Reinforcement Learning and Dynamic Discrete Choice**: The field of IRL was pioneered by (Ng et al., 2000) and has since accumulated a rich history starting with early works that rely on linear reward models and hand-crafted features (Abbeel & Ng, 2004; Ramachandran & Amir, 2007; Choi & Kim, 2011; Neu & Szepesvári, 2012). Since the introduction of the Maximum Entropy IRL (Ziebart et al., 2008; 2010) framework, the field has gravitated towards learning more flexible rewards parameterized by deep neural networks (Wulfmeier et al., 2015) paired with adversarial training (Fu et al., 2018; Ho & Ermon, 2016; Finn et al., 2016b;a) while some works attempt scale more classical methods such as Bayesian IRL (Chan & van der Schaar, 2021; Mandyam et al., 2021). Most related to our work is gradient-based apprenticeship learning (Neu & Szepesvári, 2012) which showed that the gradient of the infinite horizon Q-function in the standard RL setting satisfies a linear fixed point equation. Bellman gradient iteration (Li et al., 2017; 2019) and maximum-likelihood IRL (Vroman, 2014) take a computational approach to estimating approximate $Q$-gradients. Unlike these prior works, we generalize results to the MaxEntRL setting where we prove new properties of the Bellman score, and derive a dynamic programming algorithm that provably converges to the infinite-horizon $Q$-gradient. Most significantly, we propose a model-free score estimation algorithm which scales to high-dimensional environments and apply it to various applications beyond IRL. The field of econometrics has a rich body of work on identifying Dynamic Discrete Choice (DDC) (Rust, 1994; Arcidiacono & Miller, 2011; 2020; Abbring & Daljord, 2020) models which is equivalent to the problem of MaxEntIRL (Ziebart et al., 2008). DDC literature more so focuses on counter factual predictions (Christensen & Connault, 2019) which has various application areas such as in labor markets (Keane & Wolpin, 1994), health care (Heckman & Navarro, 2007), and retail competition (Arcidiacono & Miller, 2011).

**Imitation Learning** A related field of Imitation Learning (IL) seeks to learn policies from demonstrations (Pomerleau, 1991; Ho & Ermon, 2016; Zhang et al., 2020; Rajaraman et al., 2020; Xu et al., 2020) which is a viable alternative when the sole goal is behavior adoption. In recent years, IL has shown superior results than IRL for policy performance, particularly in the visual IL space. (Samak et al., 2021; Young et al., 2021) Techniques such as data augmentation and encoder sharing that have boosted performance in visual RL (Yarats et al., 2022) have been combined with adversarial IL methods to solve challenging control environments (Cohen et al., 2021).

## 7 CONCLUSION

We have studied the theoretical properties of the Bellman score which allowed us to derive a model-free algorithm, score-learning, for score-estimation. We showed that score-learning has various applications in IRL, behavior transfer, reward identification, and counterfactual analysis. We are looking forward to seeing future works that apply score-learning to other problems such as reward design (Hadfield-Menell et al., 2017) and explainable RL (Puiutta & Veith, 2020).

## ACKNOWLEDGEMENTS

This research was supported by NSF (#1651565), ARO (W911NF-21-1-0125), ONR (N00014-23-1-2159), CZ Biohub, and HAI. We would like to acknowledge Bertsekas & Tsitsiklis (1995) for their descriptions of the Value Iteration algorithm which sparked crucial intuitions for the theory in this work.

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
