# OpenReview forum: "Understanding and Adopting Rational Behavior by Bellman Score Estimation"
_ICLR.cc/2023/Conference — ICLR 2023 notable top 25%_

### Official Review · Reviewer_DpqQ · 2022-10-24

**Confidence:** 2
**Correctness:** 3
**Technical Novelty And Significance:** 4
**Empirical Novelty And Significance:** 4
**Recommendation:** 8

**Clarity, Quality, Novelty And Reproducibility:**

My main critics are regarding the structure of the paper, that makes difficult to guess how and why the different quantities are computed.
For ex, I had to go up to the IRL algorithm to understand why the Q-gradient is needed, and I had to go up to Sec. 4.2 to understand why the Bellman-score is needed.
Also, I still don't understand how the gradient of the reward is computed for the Q-gradient updates in Eq. 4

**Strength And Weaknesses:**

As far as I understand the maths are Ok (I haven't been through the appendix), and the approach definitely makes sense.
The results are impressive, especially for the IRL experiments with a single demonstration. The experimental protocol is correct (they used 10 random seeds for each baseline and averaged scores of policies over 20 sampled trajectories).

**Summary Of The Paper:**

This paper introduces an algorithm that computes two utile quantities, that they call Q-gradient (g(x, a)) and Bellman-score (s(x, a)), given transitions from a replay buffer, .
The Q-gradient can be used to update a reward function in IRL, and the Bellman-score to predict the adaptation of a policy to a change of the reward function.
In both cases, the experiments shows that both applications provides strong results, outperforming the baselines used for comparison.


**Summary Of The Review:**

Overall, although the paper could be more impactful and would benefit some writing clarifications, I think it already desserve communications and will benefit the RL community. I rate for a marginal accept, but may increase my score.

Minor comments:

- Preliminaries, in the unconditional occupancy measure two lines after Eq1, I think Pi_T should be replaced by Pi_0.

- At the end of page 2, I am pretty sure that the entropy as used in the MaxEntRL setting is only a function of the state, and is the entropy of the policy at a given state. Maybe a comment is missing here.

- Page 3 Line 1: "the the"

- Algo 2: "if \psi is not None" --> "if \phi is not None"

- A related work for IRL that also have strong results with single demonstrations: Primal Wasserstein Imitation Learning by Dadashi and Al.

---

> ### Author Response · Authors · 2022-11-10
> **Author Response**
>
> We thank the reviewer for taking time to carefully read our paper. We are excited that the reviewer finds our results to be strong and deserving of communication to the RL community. We hope that our response will address all of the reviewer's concerns.
>
> Q. Structure of the paper makes it difficult to know why each quantities are needed.
>
> To address the reviewer's concern, we will revise the manuscript to add descriptions of how the Q-gradient and score are going to be useful for downstream applications (e.g IRL, counterfactual predictions) at an earlier point in the manuscript, e.g the end of section 2. Please let us know if there are any other specific structural changes that the reviewer would like to see.
>
> Q. How is the gradient of the reward computed for Q-gradient updates in Eq. 4?
>
> A parametric reward function $r_{\theta}$, e.g a neural network, is learned from environment samples which enables easy computation of $\nabla_{\theta} r_{\theta}$. For most downstream applications, this component does not add overhead as learning a reward is already required. Indeed in all four of the down stream applications (reward identification, counterfactual analysis, behavior imitation, and behavior transfer) reward learning is a necessary component. We will clarify this in the manuscript.
>
> Q. Minor comments
>
> $\pi_{T}$ is correct as it is the policy executed at the 0th time-step. This is because of the backwards time indexing used in the dynamic programming literature (see footnote 1 on page 2) The expression for the conditional entropy is also correct as stated. $H_{\pi, t}(x, a)$ computes the expected sum of policy entropies at states sampled when the agent follows policy $\pi$ for $t$ time-steps starting from state-action $(x, a)$. We will revise the typos pointed out by the reviewer.

---

> > ### Comment · Reviewer_DpqQ · 2022-11-14
> > **Thanks for the clarifications**
> >
> > I think all my concerns have been addressed by the author's answer and revisions.
> > I've increased my score.

---

### Official Review · Reviewer_VmMx · 2022-10-26

**Confidence:** 4
**Correctness:** 3
**Technical Novelty And Significance:** 3
**Empirical Novelty And Significance:** 3
**Recommendation:** 5

**Clarity, Quality, Novelty And Reproducibility:**

- **Clarity:** The paper is well-written, and the flow is excellent.
- **Quality:** The quality of the paper is a bit below the standard since the background, notation, and theory are primarily based on the paper the [work](https://arxiv.org/abs/1702.08165) by Haarnoja et al. See my previous comments in the _Weaknesses_ section.
- **Novelty:** Novelty is also a bit below the standard but can still be acceptable. Nevertheless, I couldn't say this work is a high-impact study. See my previous comments in the _Weaknesses_ section.
- **Reproducibility:** The paper lacks critical information regarding reproducibility since there is no code submitted, and the experimental setup is only provided in the supplementary material. I suggest the authors provide an anonymous repository link (with an anonymous GitHub profile so that no one can identify the authors through the actual GitHub page) and briefly describe the experimental setup (e.g., hyper-parameters, implementation, etc.)


**Strength And Weaknesses:**

**Strengths:**
- The paper is well written, and the flow is excellent.
- The motivation is described well.
- Empirical studies are promising and seem to be representative of a successful comparative evaluation. For example, the proposed method achieves near-optimal performance for the IRL experiments with a single demonstration, and the use of 10 random seeds follows the widely-studied deep RL benchmarking standards.
- The approach has some novelty in it.
- The adaptations of the introduced algorithm to the considered downstream tasks seem credible.


**Weaknesses:**
- The proposed method heavily relies on the _Maximum Entropy RL (MaxEntRL)_ algorithm and the [work](https://arxiv.org/abs/1702.08165) by Haarnoja et al. (regarding notation, background, theory, etc.). While the authors claim to advance the prior art, the additional novelties are unclear and decrease the soundness of the presented work due to such an issue.
- The authors constantly discuss "model-free settings with high-dimensional continuous state-action spaces." However, the experiments are performed on a set of discrete action spaces. I couldn't understand the connection until I looked at the supplementary material in which the SAC algorithm is used for a part of the RL learner, which remains unclear in the main text.
- Can the proposed method be extended to standard continuous control (e.g., OpenAI Gym benchmarks)? I know it can't (mostly), yet the authors should address that.
- The preparation of the paper is untidy. This reduces the quality and demonstrativeness of the proposed method. See my detailed comments below.
- There are many references (maybe some belong to the supplementary text); consider reducing them. For example, the ones in the introduction section, i.e., references to the studies on economics. In addition, most of the references, e.g., the SAC algorithm, point to the pre-print versions. Please replace them with the published versions.
- I suggest the authors define the concepts of reward identification, counterfactual analysis, behavior imitation, and behavior
transfer, rather than only giving examples, as many other readers not be familiar with them.
- Although I believe that the proofs of the theorems are correct, not giving them in the main text prevents the persuasiveness of the math in the paper, which is the backbone of the proposed dynamic programming algorithm. I suggest authors provide at least a single proof in the main text that they believe is the most important.
- Although an extensive set of experiments are performed to evaluate the proposed method and compare it against the baselines, the discussion essentially translates the results in the Figures or Tables to words. Please discuss your comparative evaluations in-depth, such as why your approach outperforms baselines, what they lack, what are the limitations of your approach and, how they can be overcome, etc.

**(Very) Detailed Comments:**
- Last paragraph of section 3: There is a sentence "(We omit the finite-horizon version for brevity, but it is very similar to the infinite-horizon case)". Why didn't the authors show this (or give a reference) instead of briefly mentioning it?
- First paragraph of the introduction: The reward in an MDP may not always capture the goal. Rewards can sometimes represent the intermediate goals that are mandatory to obtain the final goal (or aim of the task). Consider changing the corresponding sentence with something like: "The reward covers the information about the goal..."
- Second paragraph of the introduction: "an RL algorithm..." not "a RL algorithm".
- Preliminaries: An MDP does not necessarily consist of a discrete state and action space. If such a case is considered, please explain why.
- First line on page 3: Duplicated "the".
- Second paragraph of Preliminaries: Consider referring to Bellman, Dynamic Programming (Dover publications) while mentioning dynamic programming.
- Line 5 of the second paragraph on page 4: Put a comma after (x, a).
- Second sentence of the first paragraph in Section 4.1: Remove "from" before the dot.
- Put floats on the same page or later when they are first mentioned. For instance, Algorithm 3 is first mentioned on page 6, but the pseudocode is given on page 5.
- Give a cross-reference to Table 1; what does it tell (e.g., "Table 1 shows the results for...")?
- I suggest the authors not highlight the highest scores in Table 2; make them bold instead.
- Put punctuation after equations if they are used in conjunction with sentences.
- Algorithm 2: "if \psi is not None" should be "if \phi is not None"


**Summary Of The Paper:**

The authors of this study make the crucial point that by understanding how variations in the underlying rewards impact optimal behavior, several discussed issues on imitation learning and inverse reinforcement learning may be resolved. This amount can be approximated locally by the Bellman score, which is the gradient of the log probability of the best strategy with respect to the reward. The authors present such a Bellman score operator, which can be used to estimate the score directly and is shown to converge to the gradient of the infinite-horizon optimal Q-values with respect to the reward. Using their theory as a guide, the authors construct an effective score-learning technique that can be applied for score estimation in high-dimensional state-actions spaces. The authors demonstrated that score-learning might be used to transfer policies between environments, accomplish state-of-the-art behavior imitation, make counterfactual predictions, and accurately identify rewards.

**Summary Of The Review:**

Given my previous detailed comments, I believe this work is consistent in terms of theory, the presentation of the introduced approach, and empirical studies. However, there are still issues with the paper. Specifically, I wouldn't say that this paper is reproducible, as no detailed experimental setup or code is given. Furthermore, the quality and novelty are slightly below the standards for a venue like ICLR. In particular, the theory and background are heavily based on prior work. Only a set of trivial calculus extends the previous approaches. Overall, I couldn't say this work is acceptable unless my concerns are addressed through straightforward comments or references.

---

> ### Author Response · Authors · 2022-11-18
> **Author Response**
>
> We thank the reviewer for taking the time and effort to read our paper. We are pleased that the reviewer found our work to be credible and our paper to be well written. Through our response and revisions, we hope to address all of the reviewer’s concerns.
>
> Q. Heavy dependence on the MaxEntRL framework and the additional novelties from Haarnoja 2017 are unclear.
>
> The works in MaxEntRL, e.g Haarnoja 2017, and ours are fundamentally tackling different problems. Their work tackles the RL problem where an agent has access to samples of the reward and seeks to learn an optimal policy. Our work tackles the problem of learning from demonstrations where samples of the optimal policy are given and we seek to learn useful quantities from them, e.g the underlying reward. The novelty of our work is that we study the Bellman score, i.e gradient of the optimal log policy probabilities with respect to the reward. We present new theoretical results on properties of the Bellman score and the score operator (Theorem 1, 2) which enables us to derive efficient, practical algorithms for computing it. (Algorithm 2) We then show that our algorithm utilizing score estimation can solve a variety of downstream tasks (reward identification, counterfactual predictions, behavior imitation, and behavior transfer) and attain state-of-the-art performance on them. The standard works in MaxEntRL, including Haarnoja 2017, do not study the Bellman score nor do they solve the four downstream tasks we presented.
>
> Q. We discuss “model-free high-dimensional continuous state-action spaces” but the experiments are performed on discrete action spaces.
>
> All of the DM control experiments for behavior imitation and reward transfer in section 5.2-5.3 are in continuous state-action spaces. We evaluated a total of 17 different control tasks (state + visual) in continuous state-action spaces. The DM control suite paper (https://arxiv.org/abs/1801.00690) contains in depth descriptions of each environment we used and their state-action spaces.
>
>
> Q. Can the proposed method be extended to standard continuous control tasks, e.g OpenAI gym benchmarks?
>
> DM Control Suite (https://arxiv.org/abs/1801.00690) contains more polished versions of essentially the same benchmark control tasks (e.g Hopper, Walker, HalfCheetah) in OpenAI gym. So our methods will certainly work on the OpenAI environments. We also note that DM Control Suite is the more up-to-date RL environment for benchmarking continuous control. For example both the PlaNet (https://arxiv.org/pdf/1811.04551.pdf) and Dreamer (https://arxiv.org/abs/1912.01603) benchmarks are based on DM control, and many prominent recent works in RL have used these benchmarks, e.g DrQ (https://arxiv.org/pdf/2004.13649.pdf), RAD (https://arxiv.org/abs/2004.14990), DrQ-v2 (https://arxiv.org/abs/2107.09645). Below we cite a paragraph from the original DM control paper which summarizes the environment.
>
> “The OpenAI Gym (Brockman et al., 2016) currently includes a set of continuous control domains that has become the de-facto benchmark in continuous RL (Duan et al., 2016; Henderson et al., 2017). The Control Suite is also a set of tasks for benchmarking continuous RL algorithms, with a few notable differences. We focus exclusively on continuous control, e.g. separating observations with similar units (position, velocity, force etc.) rather than concatenating into one vector. Our unified reward structure (see below) offers interpretable learning curves and aggregated suite-wide performance measures. Furthermore, we emphasise high-quality welldocumented code using uniform design patterns, offering a readable, transparent and easily extensible codebase. Finally, the Control Suite has equivalent domains to all those in the Gym while adding many more.”
>
> Q. Reduce references in the Introduction and replace references to preprints with the official publications
>
> Thank you for these suggestions. We have trimmed references in the main text and have replaced the preprint references with their published versions. (unless the paper is only on arxiv, e.g DM control)
>
> Q. Concretely define the concepts of reward identification, counterfactual analysis, behavior imitation, and behavior transfer.
>
> We have added these definitions to Appendix D.
>
> Q. Provide at least one most important proof in the main text.
>
> With the lower page limit of nine pages this year, it is difficult to add a full proof to the main text as it would take up at least 1.5 pages. In order to alleviate the reviewer’s concern, we can add a proof sketch of Theorem 1 in the main text in future revisions.
>
> Q. Add more in-depth analysis of comparisons to baselines in the Experiments section
>
> We have updated the experiments section to include more analysis.

---

> > ### Author Response · Authors · 2022-11-18
> > **Response Continued**
> >
> > Q. Share code for reproducibility
> >
> > For the review period, https://github.com/reviewanon/gac will be updated. The full code will be released in the next couple weeks.
> >
> > Q. Detailed Comments
> >
> > Thank you for these suggestions. We have addressed all of them in the revision. (the finite-horizon score iteration algorithm was added to Appendix D)

---

> > > ### Comment · Reviewer_VmMx · 2022-11-21
> > > **Thank you for response but further action is needed**
> > >
> > > **Response of the authors:**
> > > I would like to thank the authors for their detailed responses. I am currently satisfied with their study, and the novelty and contributions are now clearer. If the current page limit is still an issue, I think you don't need to put a sketch proof. Other than these, I still have minor issues outlined below.
> > >
> > > **Current issues:**
> > > - Can the authors highlight the changes made, for example, by the color red? So that I can easily notice the revised parts. Just define ``\newcommand{\review}[1]{\textcolor{red}{#1}}``, and put text in ``\review{[some text]}``. Then, it will print red text. You can change the option to ``\textcolor{black}{#1}`` in the camera-ready version.
> > > - Please upload the code, the repository raises a 404 error.
> > >
> > > Please do these ASAP so that I can increase my score.

---

> > > > ### Author Response · Authors · 2022-12-11
> > > > **Author response continued**
> > > >
> > > > We thank the reviewer for their enthusiastic engagement and additional feedback. We apologize for the delay in response. We believe there were issues with the open review email notification system as the first author was not receiving notifications about reviewer comments posted after the rebuttal period. Unfortunately, we are unable to highlight the changes made as revision uploads are closed after the rebuttal period. However, we have summarized the location of the changes below to facilitate the reviewer's reading. Regarding the broken code link, we will fix it ASAP within the next 24 hours and follow-up.
> > > >
> > > > Section 1 - Reduced references
> > > >
> > > > Last sentence of results paragraph in section 5.1, last two sentences of section 5.1, last paragraph of section 5.2, last sentence of section 5.3 - Added more in-depth experimental analysis.
> > > >
> > > > References - replaced preprints with journal versions.
> > > >
> > > > Appendix D.2 - Added detailed problem definitions of reward identification, counterfactual analysis, behavior imitation, and behavior transfer.
> > > >
> > > > All reviewer's suggested revisions in the "(Very) Detailed Comments" section were addressed in the locations pointed out by the reviewer.

---

> > > > ### Author Response · Authors · 2022-12-12
> > > > **Code link fixed**
> > > >
> > > > We thank the reviewer for their patience. We have created a new repo to fix the broken code link. Please find the code here: https://github.com/reviewanon/gac_temp

---

> > > > ### Author Response · Authors · 2022-12-14
> > > > **One more fix to the code**
> > > >
> > > > We discovered that in the codebase push made two days ago, we had mis-specified the reward learning rate in the hyper-parameter file which will not generate the correct results. We have just pushed the changes with the corrected hyper-parameters in the config file.

---

### Official Review · Reviewer_nKZi · 2022-10-28

**Confidence:** 3
**Correctness:** 4
**Technical Novelty And Significance:** 4
**Empirical Novelty And Significance:** 2
**Recommendation:** 8

**Clarity, Quality, Novelty And Reproducibility:**

Clarity: a few details are missing (see comments), the paper is quite dense, otherwise clear

Quality: high

Novelty: novel and very interesting idea

Reproducibility: code is missing, the appendix contains the hyperparameters for GAC


**Strength And Weaknesses:**

### Strengths

This is as far as I can tell a novel, neat idea that is quite versatile.

No knowledge of the environment dynamics is required.

The paper is overall well-written, clear, and to the point. It is a bit dense though, and might benefit from additional figures.

Empirical results on various tasks (IRL, imitation learning, reward transfer) are good.

### Weaknesses

I would advocate for using optimal (or near-optimal) instead of the term rational that I find confusing, e.g. one could think that it means sensible according to human evaluation. Unless this is related to the fact of using demonstrations, in which case it should be better explained.

Nit: Algorithm 1 misses the initialization of the Bellman score

It is not clear to me how the gradient of the reward appears in the Bellman score operator (definition 2). Could authors expand on that and add a clarification in the text?

Algorithm 2 leverages the gradient of the reward. Does that mean that score learning only works in tasks where a reward parameterization exists and is known? In that case, I think the current state of the paper is not clear enough about this. I do get that this is not a limitation for IRL and IL tasks since a reward is to be learned anyway.

I encourage the authors to open-source their code to maximize the impact of their contribution.


**Summary Of The Paper:**

The paper introduces the Bellman score, which is the gradient of the log-probabilities of the optimal policy with respect to the reward parameters, and a corresponding operator to learn the gradient of the Q-function (which is related to this quantity) via fixed-point iteration.
The authors show that score-learning is a versatile approach and derive an actor-critic algorithm for IRL (GAC).
They empirically study several applications: imitation learning, inverse RL, reward transfer, counterfactual estimation.

**Summary Of The Review:**

This is an exciting paper with a lot of potential impact.

I have made a small number of questions/comments that could be addressed by the authors.

I recommend acceptance.

---

> ### Author Response · Authors · 2022-11-08
> **Author Response**
>
> We thank the reviewer for their valuable time and effort into reading our paper. We are happy that the reviewer finds our work to be versatile and impactful. We hope that our response will clarify any misunderstandings and address all of the reviewer's concerns.
>
> Q. Using "optimal" instead of "rational"
>
> In paragraph one we described how rational behavior is characterized as reward maximizing behaviors for MDPs, i.e optimal policies. We agree with the reviewer that the term “optimal policy” is more precise and mathematically formal than “rational behavior”, and thus it is used throughout the rest of the paper after page one. We sought to start with the more familiar term “rational behavior” in order to give more intuition for how optimal policies relate to the behavior in the real world, then transition to using formal terms throughout the rest of the paper.
>
> Q. Not clear how the gradient of the reward appears in the Bellman score operator (def 2)
>
> The Bellman score is the gradient of the log optimal policy probabilities with respect to the underlying reward (def 1). The score operator (def 2) computes this quantity by essentially aggregating the per time-step reward gradients by a dynamic programming procedure. When the reward is fully tabular, the gradient of the reward is simply the identity matrix as explained in the paragraph under def 2.
>
> Q. Score learning only works when the reward parameterization exists and is known?
>
> The current version of the theory is indeed derived for parametric reward families. As the reviewer mentioned, this is not an issue for IRL since we seek to learn a parametric reward. We believe it's be possible to extend our results to non-parametric rewards, where the score operator would most likely involve functional gradients, but leave this to future work. Following the reviewer's advice, we will clarify this point in the paper.
>
> Q. Open sourcing code
>
> We will be open sourcing shortly.

---

> > ### Comment · Reviewer_nKZi · 2022-11-17
> > **Response to authors' response**
> >
> > I want to thank authors for their reply to my comments, which clarified several elements.

---

### Official Review · Reviewer_9iGT · 2022-10-30

**Confidence:** 5
**Correctness:** 2
**Technical Novelty And Significance:** 3
**Empirical Novelty And Significance:** 2
**Recommendation:** 5

**Clarity, Quality, Novelty And Reproducibility:**


Clarity: Overall, this paper is well-writen and is easy to follow.

Novelty: The authors propose the concepts of Bellman score and Q-gradient, which are novel to my knowledge. The algorithm for learning Q-gradient is similar to DQN and thus is not very novel.

**Strength And Weaknesses:**


Strength:

1. The authors propose the concepts of Bellman score and Q-gradient, which are novel to my knowledge. The proposed Bellman score and Q-gradient have an important role on the problems of understanding and learning optimal policies from demonstrations, including inverse reinforcement learning (IRL) and counterfactual analysis.





Weakness:

1. In section 4.1, they derive a new formulation for the gradient of the MaxEntIRL (i.e., Eq. (7)) and propose a new MaxEntIRL algorithm GAC based on Eq. (7) and Score-learning. However, I cannot see the benefits of GAC over existing IRL methods like GAIL, DAC.

   The authors claim that existing IRL methods utilize the formulation of Eq. (8) and thus suffer the issues of data reusing and high variance. First, for the data reusing issue, DAC has already addressed this issue by using data in replay buffer to update rewards.

   Second, I think there is no high variance issue in performing SGD updates based on Eq. (8) and the analogy between Eq.(8) and MC-PG method made by the authors does not make sense. In Eq. (8), we use the sample average of reward gradients to estimate the expectation. Unlike the policy value (a multi-step value with a scale of 0 to T) estimated in MC-PG method, the reward gradient is a one-step value and thus using MC estimates does not have a high variance. Instead, GAC based on Eq. (8) could have a high variance since it only uses initial state-action pairs for estimating.

2. The computational cost of GAC is large. Compared with existing IRL methods like GAIL, AIRL and DAC, GAC additionally maintains the Q-gradient network, target Q-gradient network and score network. As pointed in Appendix C, GAC requires roughly 2+ more compute time than the baselines.

3. The authors claim that score learning can be applied in behavior transfer. In experiments, they show that learned rewards from GAC can transfer behaviors to perturbed environments. As far as I am concerned, unlike Adversarial IRL, GAC does not contain special algorithmic designs for handling transfer. It remains a question why GAC can transfer rewards to new environments.

4. The theory in this paper is derived within the scope of MaxEntRL. The authors claim that similar results for the standard RL can be found in the Appendix. However, I do not see these results in the Appendix.

5. This paper contains the review of algorithm progress in IRL and IL. However, some basic works on the theory of IRL and IL are missing, see e.g., [1, 2, 3].

References:

[1] Y. Zhang, Q. Cai, Z. Yang, and Z. Wang. Generative adversarial imitation learning with neural network parameterization: Global optimality and convergence rate. ICML, 2020.

[2] N. Rajaraman, L. F. Yang, J. Jiao, and K. Ramchandran. Toward the fundamental limits of imitation learning. NeurIPS, 2020.

[3] T. Xu, Z. Li, and Y. Yu. Error bounds of imitating policies and environments. NeurIPS, 2020.

Minor issues:

1. The first line in page 3: "the the soft Q-values"
2. In definition 3, the shape of the Bellman score should be d \times |X \times A|?

3. In the line of "update Q-gradient" in Algorithm I, g (x, a) should be g (x', a') under the transition probability of P (x', a' |x, a).


**Summary Of The Paper:**

This paper focuses on the problems of understanding and learning optimal policies from demonstrations, including reward identification, counterfactual analysis, behavior imitation and behavior transfer. Under the  framework of MaxEntRL, they first propose the concept of Bellman score which captures how changes in the reward functions influence the optimal policies, and further show that Bellman score can be calculated as a difference of the gradients of optimal Q-values w.r.t the reward parameters. Then they introduce the Bellman score operator and prove that applying this operator repeatedly yields the true Q-gradients. Based on this operator, they develop a model-free algorithm Score-learning to learn Bellman score with finite samples. Next, they apply the Score-learning method on tasks of IRL and counterfactual predictions. In experiments, they evaluate that applying score-learning can achieve SOTA performance on the mentioned four tasks.

**Summary Of The Review:**

In this work, the authors propose the concept of Bellman score, which is novel. To learn Bellman score with finite samples, they develop the algorithm score learning based on DQN. The Score-learning method has a broad applicability on tasks of IRL and counterfactual predictions. However, the benefits of the developed IRL method GAC over existing IRL methods are not clear. Besides, GAC suffers the computational issue. Some claims in this paper are not well supported. In conclusion, I recommend the score of 5.

---

> ### Author Response · Authors · 2022-11-18
> **Author Response**
>
> We thank the reviewer for their valuable time and effort put into reading our paper. We are happy that the reviewer has found our work to be novel and broadly applicable. Through our response and revisions, we’d like to address all of the reviewer’s feedback and concerns.
>
> Q. Advantage of GAC over IRL methods such as GAIL or DAC?
>
> DAC and GAIL are Imitation Learning (IL) methods (not IRL methods) that recover a discriminator which does not correspond to the true reward function, but rather a saddle point solution to the min-max optimization problem $max_{\pi} min_{D} E_{\pi}[log D(x, a)] + E_{\pi^{*}}[log (1 - D(x, a))]$. (As noted at the end of section 3 in the DAC paper) Thus simply maximizing the learned discriminator from scratch will not lead to an optimal policy that matches the demonstrations. In contrast, we seek to solve the IRL problem where recovering true reward is the primary goal. Learning the reward has several advantages over IL methods since the recovered reward can be used for a variety of downstream applications, such as reward transfer and counterfactual predictions, in addition to behavior imitation. The inferred reward itself also offers valuable insight into interpreting the incentives of rational decision makers when IRL is used for behavioral modeling in econometrics or neuroscience. (Introduction contains concrete examples) Furthermore, our results show that, empirically, GAC attains superior behavior imitation performance compared to pure IL methods such as DAC.
>
> Q. DAC has already addressed the data reuse issue by using a replay buffer
>
> If we directly apply the replay buffer technique from DAC to Eq. 8 by replacing the on-policy samples by replay buffer samples, this objective does not recover the original expectation as the importance weights are omitted. (stated at the end of section 4 in the DAC paper) In contrast, our key idea is to use Q-learning which implicitly accounts for the importance weights of the changing policy while also being able to reuse old samples in the replay buffer. We posit that this may be the key reason GAC outperforms DAC on the behavior imitation experiments.
>
> Q. There are no high variance issues with Eq. 8. Unlike GAC, MC-PG seeks to estimate a multistep value (with horizon T) which causes high variance.
>
> We believe this claim is wrong. The second term of Eq. 8, i.e $E_{\rho_{\pi, T}}[\nabla_{\theta} r_{\theta}(x, a)]$, is also a multi-step value as it is sampled from the (unnormalized) occupancy measure $\rho_{\pi, T}(x, a) = \sum_{t = 0}^{T} \gamma^{t} p_{\pi, t}(x, a)$ which is the sum of marginals across multiple time-steps. We could equivalently write the second term of Eq. 8 as $E_{\pi}[\sum_{t = 0}^{T} \gamma^{t} \nabla_{\theta} r_{\theta}(x_t, a_t)]$. The multistep sum is just hidden in the occupancy measure. In fact, if we consider a one-dimensional parameter $\theta$, the second term of Eq. 8 is precisely the multi-step value in MC-PG for a reward function defined as $\tilde{r}(x, a) = \nabla_{\theta} r_{\theta}(x, a)$, i.e $E_{\pi}[\sum_{t = 0}^{T} \gamma^{t} \tilde{r}(x_t, a_t)] = E_{\pi}[\sum_{t = 0}^{T} \gamma^{t} \nabla_{\theta} r_{\theta}(x_t, a_t)]$. Both the second term of Eq. 8 and the MC-PG objective suffer from high-variance as they are taking samples from a multi-timestep (unnormalized) distribution $\rho_{\pi, T}(x, a)$ which is likely to be diffuse (as the policy visits a wider range of different states when given more time to act in the environment).
>
> Q. Computational cost of GAC is high
>
> When there is no weight sharing between the different modules, e.g in state-control tasks, the computational cost of GAC is roughly $2 \times$ higher than the baselines as we stated in Appendix C. However, when there is weight sharing between the different learnable components we found that the computational cost of GAC is significantly reduced. In recent visual control experiments, we found that the encoder, updated with only the critic loss, can be shared across all modules — i.e actor, critic, Q-gradient, and reward — without compromising performance. As the cost of updating the encoder comprises a majority of the computational time, the overall training time of GAC was only $1.2 \times$ higher than the baselines on average across all visual control tasks. We have added these new results to Appendix D. We also note that the score network does not have to be learned for the purposes of IRL, so it should not be included in the compute time comparison against GAIL, AIRL, and DAC.

---

> > ### Author Response · Authors · 2022-11-18
> > **Author Response Continued**
> >
> > Q. Why can GAC handle reward transfer when there are no special algorithmic designs for it?
> >
> > The relevant theoretical result of AIRL states that a disentangled reward (i.e transferrable across dynamics) must be a state-only reward (Theorem 5.2 in the AIRL paper). Thus the relevant algorithmic design that AIRL incorporates is to learn a state-only reward. (The potential shaping terms aren’t designed to guarantee reward transfer, but rather to mitigate effects of unwanted shaping) For this reason, GAC also learns a state-only reward function for the transfer experiments as stated in Appendix B (baselines section). To make this point more clear, we have included this explanation of why a state-only reward was learned to Appendix B.
> >
> > Q. Theory in the case for standard RL is missing in the Appendix
> >
> > Thank you for catching this. We missed removing the sentence regarding standard RL results from the manuscript. An earlier draft of our work included sub-gradient methods for the case of standard RL, but we removed it as the theory did not add much practical value in guiding experiments. The sentence has been removed in the revision.
> >
> > Q. Additional references for IRL and IL.
> >
> > Thank you for these suggestions. We have added these references to the manuscript.

---

> > ### Comment · Reviewer_9iGT · 2022-11-25
> > **Most of my concerns are not addressed**
> >
> > Thank the authors for their response. Most of my concerns are not addressed. Below are my further comments.
> >
> >
> >
> > **DAC and GAIL are Imitation Learning (IL) methods (not IRL methods) that recover a discriminator which does not correspond to the true reward function, but rather a saddle point solution to the min-max optimization problem. In contrast, we seek to solve the IRL problem where recovering true reward is the primary goal.**
> >
> >
> >
> > Comment:
> >
> > I think this claim is wrong. First, at the formulation level, both GAIL/DAC and MaxEnt IRL/GAC are IRL methods based on occupancy measure matching. IRL is a dual of an occupancy measure matching problem and the recovered cost function is the dual optimum [1].
> >
> > Specifically, GAC builds on the MLE objective of MaxEnt IRL (Eq. (6)). Importantly, there exists an equivalence between the MLE objective and the dual problem induced by occupancy measure matching (see Lemma 3.3 in [1]). Besides, GAIL/DAC is also under the framework of occupancy measure matching.
> >
> > Second, at the algorithmic level, both GAIL/DAC and MaxEnt IRL/GAC solve a min-max problem. In the min-max problem, the policy aims to maximize the current reward (the RL step in algorithm 2), while the reward aims to maximize the expert's value but minimize the policy's value (the reward update in algorithm 3). Thus, GAIL/DAC does not differ from MaxEnt IRL/GAC in terms of solving a min-max problem from the algorithmic perspective.
> >
> >
> >
> > **The gradient estimator based on Eq. 8 resolves the high variance issue existed in methods that utilize the gradient estimator based on Eq. 7. **
> >
> > Comment:
> >
> > In my opinion, the gradient estimator built by Eq. 7 has a higher variance than that based on Eq. 8. This is because the gradient estimator based on Eq. 7 only uses the collected initial states while the number of initial states is extremely small. This claim can be supported by many existing works like ValueDICE [3], IQL [4]. Similar to this work, ValueDICE and IQL also derive a gradient estimator that uses initial states and Q-gradient, which is similar to Eq. 8. In IQL (subsection 5.1), they show that optimizing in initial states results in instability for convergence. To address this issue, they utilize a gradient estimate similar to Eq. 8 in the online setting.
> >
> >
> >
> > Based on the above two comments, I do not clearly see the benefits of GAC over existing works like GAIL, DAC, IQL.
> >
> >
> >
> > **Following AIRL, GAC also learns a state-only reward function for the transfer experiments as stated in Appendix B**
> >
> > Comment:
> >
> > Learning a state-only reward function is not sufficient for solving the reward transfer tasks. Theorem 5.2 in the AIRL paper only suggests the necessity of learning a state-only reward under the condition that the ground-truth reward is also state-only.
> >
> >
> >
> > **Extension to standard RL**
> >
> > Comment:
> >
> > I think this extension is important. The proposed method in this paper is restricted to the MaxEnt RL setting since the derivation largely depends on the nice property that the optimal policy can be formulated as the softmax of the optimal Q-function. In this standard RL setting, this nice property does not hold. It is unknown whether the current method can be extended to the standard RL setting.
> >
> >
> >
> > References:
> >
> > [1] Generative Adversarial Imitation Learning
> >
> > [2] A Primer on Maximum Causal Entropy Inverse Reinforcement Learning
> >
> > [3] IQ-Learn: Inverse soft-Q Learning for Imitation
> >
> > [4] IMITATION LEARNING VIA OFF-POLICY DISTRIBUTION MATCHING

---

> > > ### Author Response · Authors · 2022-12-11
> > > **Author Response continued**
> > >
> > > We again apologize for the delay in response. We thank the reviewer for their enthusiastic follow-up and would like to fully address the reviewer's remaining concerns.
> > >
> > > Q. GAIL/DAC and GAC are both operating under the occupancy matching framework since MaxEntIRL is its dual problem.
> > >
> > > We agree with this statement, but it is orthogonal to our point which is that the discriminator learned in IL methods such as GAIL/DAC do not recover the true reward function unlike IRL methods such as GCL and GAC. Thus, IRL methods such as GAC and GCL have the additional advantage of learning the reward which may be used for a variety of downstream tasks such as behavior transfer and counterfactual predictions.
> > >
> > > As stated by the reviewer, the essence of the GAIL theory is that the policy learned by RL on the cost recovered by MaxEntIRL is equivalent to the policy learned by occupancy matching with an entropy regularizer. (Eq. 4 of [1]) This statement, however, **is not equivalent** to the claim that the discriminator learned in the process of occupancy matching for GAIL recovers the true reward. This claim is in fact false. When the learner policy imitates the expert perfectly, the GAIL discriminator outputs a constant for all state-action pairs. Other works have also recognized this fact, e.g last paragraph of section 3 in the AIRL paper [2] states, "GAIL does not place special structure on the discriminator, so the reward cannot be recovered".
> > >
> > > Q. GAIL/DAC and GAC are algorithmically similar in that they both solve a min-max optimization problem.
> > >
> > > While the overall algorithmic structure of applying coordinate descent to a min-max objective is similar, the key differences are the parameterization of the discriminator and the loss functions which lead to significant differences in what quantities are estimated. The discriminator $D$ in GAIL minimizes the loss $E_{\pi}[log D(x, a)] + E_{\pi^*}[log (1 - D(x, a))]$ while GAC and GCL minimize $E_{\pi}[D(x, a)] - E_{\pi^*}[D(x, a)]$. We re-emphasize that these algorithmic differences are non-trivial as they determine whether or not the true reward can be estimated. As mentioned before, the AIRL paper [2] also points out that GAIL cannot recover the true reward because it does not place any special structure on the discriminator. As a side note, AIRL shows two different special discriminator structures which enables estimation of either (1). the advantage reward (2). the true state-only reward in the case that the dynamics satisfies a set of rather restrictive disentanglement assumptions. On the other hand, GAC and GCL are able to more generally estimate the true state-action dependent rewards (not just the advantage reward) within equivalence classes dependent on identifiability properties of the environment, e.g up to an additive constant in the best case scenario.
> > >
> > > Q. Gradient estimator in Eq. 7 has higher variance than Eq. 8 because it only uses the collected initial states.
> > >
> > > For variance comparisons, the estimator derived from Eq. 7 clearly has lower variance than Eq. 8 as we have explained in our previous response that the comparison between Eq. 7 and 8 is precisely equivalent to the bias-variance trade-off between Monte-Carlo policy gradient and Actor-critic policy gradient methods. Regarding the issue of updating on a limited number of initial states, we would like to clarify that the process of estimating the Q-gradient $g_{\phi}$ uses all state-action pairs collected from the environment and not just the initial states, i.e we take samples $(x, a, x’, a’)$ from the replay buffer to minimize $(\nabla_{\theta} r_{\theta}(x, a) + \gamma g_{\phi}(x’, a’) - g_{\phi}(x, a))^{2}$. The reward is updated by following the Q-gradient evaluated at the initial states which is an accumulation of all the gradients at state-action pairs visited afterwards. Thus, the reward is also updated on all visited state-actions and not just the initial states. This is different from the overfitting issue pointed out in the IQL and ValueDICE paper where the module being estimated, i.e value function, is only optimized on the initial states. We would also like to emphasize that Eq. 7 works well in practice as shown by our experiments.

---

> > > > ### Author Response · Authors · 2022-12-11
> > > > **Author Response continued**
> > > >
> > > > Q. Learning a state-only reward function is not sufficient for solving the reward transfer tasks.
> > > >
> > > > We agree that it is not a sufficient condition, but this is orthogonal to the point we were making. Learning a state-only reward is the main algorithmic design of the AIRL paper which helps to handle behavior transfer. (section 6 of AIRL paper) As the reviewer had previously said GAC does not have algorithmic designs to help with transfer, we were pointing out that GAC in fact has the same core algorithmic design as AIRL for facilitating transfer which is to parameterize the reward as state-only.
> > > >
> > > > Q. Extension to standard RL would be important
> > > >
> > > > We believe MaxEntRL is a flexible framework that's used across many prominent works in RL, e.g SAC and DrQ, and also other fields such as econometrics. It has the benefit of of characterizing the optimal behavior in many realistic scenarios where agents operate in the presence of observable perturbations (shocks) to the reward function. [4]
> > > > Additionally, MaxEntRL provides a framework that naturally facilitates exploration which accelerates the RL process. We would also note that the theoretical justifications in the reviewer's cited references GAIL, DAC, IQL all rely on the MaxEntRL framework. While we agree that an extension to standard RL would be nice, we do not think operating in the MaxEntRL framework is a strong limitation of our work.
> > > >
> > > > [1] Generative Adversarial Imitation Learning, Ho et al.
> > > >
> > > > [2] Adversarial Inverse Reinforcement Learning, Fu et al.
> > > >
> > > > [3] On Contrastive Divergence Learning, Carrerira-Perpinan et al.
> > > >
> > > > [4] Structural Estimation of Markov Decision Processes, Rust et al.

---

> > > > > ### Comment · Reviewer_9iGT · 2022-12-11
> > > > > **Response to the authors**
> > > > >
> > > > > Thank the authors for their response. My concern on the argument about the variance of Eq.(7) is addressed. Below are my further comments.
> > > > >
> > > > >
> > > > >
> > > > > **Advantage of GAC over existing IL methods DAC/GAIL and IRL methods AIRL/GAC**
> > > > >
> > > > > The current discussion deviates from my original concerns. My original question is what is the advantage of GAC  based on the new scoring learning procedure over existing IL methods DAC/GAIL and IRL methods AIRL/GCL. The authors claimed that compared with GAIL/DAC, the significant algorithmic differences are the parameterization of the discriminator and the loss functions. However, these algorithmic designs have been utilized in existing methods AIRL/GAC. Thus I am still not aware of the advantage of GAC.
> > > > >
> > > > >
> > > > >
> > > > > As a minor point, the authors argued that the key differences in the parameterization of the discriminator and the loss functions determine whether or not the true reward can be estimated. I think this argument is not fully correct. The loss function does not determine whether or not the true reward can be estimated. GAIL minimizes the JS divergence which results in the discriminator loss of $E_\pi[\log D(x, a)]+E_{\pi^*}[\log (1-D(x, a))]$ while GAC/GCL can be regarded to minimize the IPM distance which  leads to the loss of $E_\pi[D(x, a)]-E_{\pi^*}[D(x, a)]$.
> > > > >
> > > > >
> > > > >
> > > > > Moreover, if my understanding is correct, as shown in the second paragraph in appendix B, the reward $r_\theta$ in GAC does not utilize a special discriminator structure as in AIRL and GCL.
> > > > >
> > > > >
> > > > >
> > > > > **GAC in fact has the same core algorithmic design as AIRL for facilitating transfer which is to parameterize the reward as state-only**
> > > > >
> > > > > This is exactly the main issue. This paper does not propose its own algorithmic components for reward transfer tasks. For reward transfer tasks, GAC and AIRL have the same algorithmic designs.  So for reward transfer tasks, what is the advantage of GAC over AIRL in the level of algorithmic designs? Moreover, GAC is empirically better than AIRL for reward transfer tasks. It is still unclear why GAC can have these empirical gains. Therefore, I recommend removing the results on reward transfer since the unique algorithmic design proposed in this paper (score learning) is largely orthogonal to reward transfer.

---

> > ### Comment · Reviewer_9iGT · 2022-12-08
> > **concerns are not addressed**
> >
> > As the authors overlooked or avoided my unsolved concerns (see https://openreview.net/forum?id=WzGdBqcBicl&noteId=dGj-C5HMT5R).
> >
> > I still keep my opinion this work is marginally below the acceptance threshold and consider declining my score in recent days.

---

> > ### Comment · Reviewer_9iGT · 2022-12-10
> > **The score dropped**
> >
> > As the authors were blind to the concerns (https://openreview.net/forum?id=WzGdBqcBicl&noteId=dGj-C5HMT5R) I raised, I dropped my score (which may change according to the response).
> > And I sincerely suggest other reviewers and AC reexamine my concerns and the paper.
> >
> > Yours,
> > Reviewer 9iGT

---

> > > ### Author Response · Authors · 2022-12-11
> > > **Apologies for delayed response**
> > >
> > > We apologize for the delay in response. We believe there were issues with the open review email notification system as the first author was not receiving notifications about reviewer comments posted after the rebuttal period. The first author saw the reviewer's additional comments today after another author informed them about the updates. Please see our response to the reviewer's additional comments below.

---

> > ### Comment · Reviewer_9iGT · 2022-12-11
> > **Response to the authors**
> >
> > Thank the authors for their response. My concern on the argument about the variance of Eq.(7) is addressed. Below are my further comments.
> >
> >
> >
> > **Advantage of GAC over existing IL methods DAC/GAIL and IRL methods AIRL/GAC**
> >
> > The current discussion deviates from my original concerns. My original question is what is the advantage of GAC  based on the new scoring learning procedure over existing IL methods DAC/GAIL and IRL methods AIRL/GCL. The authors claimed that compared with GAIL/DAC, the significant algorithmic differences are the parameterization of the discriminator and the loss functions. However, these algorithmic designs have been utilized in existing methods AIRL/GAC. Thus I am still not aware of the advantage of GAC.
> >
> >
> >
> > As a minor point, the authors argued that the key differences in the parameterization of the discriminator and the loss functions determine whether or not the true reward can be estimated. I think this argument is not fully correct. The loss function does not determine whether or not the true reward can be estimated. GAIL minimizes the JS divergence which results in the discriminator loss of $E_\pi[\log D(x, a)]+E_{\pi^*}[\log (1-D(x, a))]$ while GAC/GCL can be regarded to minimize the IPM distance which  leads to the loss of $E_\pi[D(x, a)]-E_{\pi^*}[D(x, a)]$.
> >
> >
> >
> > Moreover, if my understanding is correct, as shown in the second paragraph in appendix B, the reward $r_\theta$ in GAC does not utilize a special discriminator structure as in AIRL and GCL.
> >
> >
> >
> > **GAC in fact has the same core algorithmic design as AIRL for facilitating transfer which is to parameterize the reward as state-only**
> >
> > This is exactly the main issue. This paper does not propose its own algorithmic components for reward transfer tasks. For reward transfer tasks, GAC and AIRL have the same algorithmic designs.  So for reward transfer tasks, what is the advantage of GAC over AIRL in the level of algorithmic designs? Moreover, GAC is empirically better than AIRL for reward transfer tasks. It is still unclear why GAC can have these empirical gains. Therefore, I recommend removing the results on reward transfer since the unique algorithmic design proposed in this paper (score learning) is largely orthogonal to reward transfer.

---

> > > ### Author Response · Authors · 2022-12-12
> > > **Author response continued**
> > >
> > > We thank the reviewer for their response. We are glad that the reviewer's concerns are being resolved and answer the remainder of their questions below.
> > >
> > > Q. Advantage of GAC over existing IL methods DAC/GAIL and IRL methods AIRL/GCL
> > >
> > > In general, the use of an actor-critic style algorithm, i.e score-learning, to estimate the Q-gradient which updates the reward is a key algorithmic difference that distinguishes GAC from all baselines DAC/GAIL/AIRL/GCL. We detail more specific comparisons with each baselines below. We can also add these descriptions to the camera-ready version if the paper is accepted.
> > >
> > > Compared to DAC/GAIL the key differences are (1). GAC can estimate the true reward while DAC/GAIL cannot (the discriminator will output a constant at convergence), (2). They differ in their choice of loss function, i.e DAC/GAIL discriminator minimizes the GAN objective $E_{\pi}[log D(x, a)] + E_{\pi^*}[log (1 - D(x, a))]$ while the GAC discriminator (reward) optimizes the maximum likelihood objective $E_{\pi^*}[-log (e^{D(\tau)} / Z)]$, (3). GAC re-uses data in a principled manner with a Q-learning style algorithm, i.e score-learning, to learn the Q-gradient network while GAIL only uses recent on-policy samples and DAC uses replay buffers samples with the importance weights omitted and thus cannot recover the true expectation in its discriminator objective. (stated at the end of section 4 in the DAC paper)
> > >
> > > Compared to AIRL the key differences are (1). AIRL uses a particular parameterization of the discriminator which restricts the space of recoverable rewards to either the advantage reward or the state-only reward when the dynamics satisfies disentanglement assumptions (2). GAC is able to reuse old data while AIRL only uses most recent on-policy samples.
> > >
> > > Compared to GCL, the key differences are (1). GAC reuses data with a Q-learning style algorithm to reuse old data while GCL uses importance-sampling (the latter is computationally heavy as it requires saving a history of policies), (2). GAC uses an actor-critic style algorithm (score-learning) to update the reward while GCL uses Monte-Carlo (MC) samples (the former trades off unbiasedness for lower-variance, and we empirically demonstrate that this trade-off improves IRL performance similar to how actor-critic RL algorithms typically outperform MC policy gradient methods)
> > >
> > > In addition to these differences, we would like to emphasize that score-learning has applications in addition to IRL. We are able to compute the Bellman score which enables counterfactual predictions (a core application of interest in fields such as Econometrics).
> > >
> > > Q. The loss function does not determine whether the true reward can be estimated
> > >
> > > We made a typo in our previous response. The objective of GAC/GCL is not the IPM distance but the maximum likelihood objective, $E_{\tau \sim \pi^*}[-log (e^{D(\tau)} / Z)]$. This is a different loss function from that of GAIL/DAC in the sense that it minimizes KL instead of JS divergence and the distributions being compared are the trajectory distributions instead of the occupancy measures. The choice of loss function alone does not fully determine what quantities are able to be estimated, but the combination of the loss and discriminator parametrization do. For example, AIRL combines the GAN loss with a special discriminator parameterization which enables estimation of the advantage reward while GAIL omits the parametrization and cannot estimate the reward. On the other hand, GAC/GCL optimizes the MLE loss without a special discriminator structure and is still able to estimate the true reward.
> > >
> > > Q. GAC does not have a unique algorithmic design to handle reward transfer
> > >
> > > We believe GAC is able to outperform AIRL for reward transfer tasks because it is able to more accurately estimate the state-only reward. As the accuracy of the state-only reward is important for how useful it is when optimized in a perturbed environment, better reward estimation performance subsequently leads to better transfer performance. Although GAC does not have additional algorithmic designs for reward transfer, we believe it is still valuable to quantify its performance on the reward transfer task as it is a core application area of IRL.
> > >
> > >
> > > [1]. A Connection Between Generative Adversarial Networks, Inverse Reinforcement Learning, and Energy-Based Models
> > >
> > > [2]. Discriminator-Actor-Critic: Addressing Sample Inefficiency and Reward Bias in Adversarial Imitation Learning

---

### Official Review · Reviewer_9PVy · 2022-10-30

**Confidence:** 4
**Correctness:** 4
**Technical Novelty And Significance:** 3
**Empirical Novelty And Significance:** 3
**Recommendation:** 8

**Clarity, Quality, Novelty And Reproducibility:**

The paper is quite clearly written and the work seems to be of high quality. I have not complaints about writing and exposition. The underlying idea of gradient-based IRL is not new, but this paper makes basic yet fundamental observations about the structure of the gradient and as a result is able to formulate substantially more efficient algorithms for its estimation.

**Strength And Weaknesses:**

This paper has a strong basic insight, which is utilized well. There are no obvious flaws that struck me while reading.

**Summary Of The Paper:**

This paper contributes to the study of gradient-based inverse reinforcement learning, in particular, learning a reward function by gradient ascent, under which an observed policy is optimal. The main problem of interest is the estimation of the gradient of the policy return with respect to the parameters of the reward function. It is shown that this gradient follows a dynamic-programming Bellman-type equation similar to the actual policy return, which can then be exploited to derive gradient estimation algorithms based existing RL algorithms. Substantially improved performance is demonstrated on benchmarks.

**Summary Of The Review:**

This paper makes an important, well-argued contribution to a relevant problem. Novelty and significance are relatively high.

---

> ### Author Response · Authors · 2022-11-08
> **Author Response**
>
> We greatly appreciate the reviewer's appreciation of our work. We are happy that the reviewer found our paper to have strong insights and  well-argued contributions that are significant. Please let us know if there are any points we can address.

---

### Official Review · Reviewer_1Fw8 · 2022-10-31

**Confidence:** 3
**Correctness:** 4
**Technical Novelty And Significance:** 4
**Empirical Novelty And Significance:** 3
**Recommendation:** 8

**Clarity, Quality, Novelty And Reproducibility:**

The paper was pretty easy to read. As discussed above, the results were clearminded and novel.

**Strength And Weaknesses:**

Strengths: this paper demonstrates strong benefits of using the Bellman score, in particular for maximum entropy IRL and counterfactual predictions. And most crucially, they discover a dynamic programming algorithm that enables its efficient computation. The algorithm is conceptually solid and reasonable, and well motivated by classic ideas in deep RL (esp DQN, bootstrapping, etc). The demonstrations on IRL and counterfactuals were strong. Relevant baselines were compared, and a good variety of tasks were used.

Overall, for its purpose, it is pretty solid. Develops theory and shows relevant experiments to make a central coherent point about the benefits of efficiently computing Bellman score.


Weaknesses: More details on exact implementation would be helpful, so that readers can have a chance to replicate and further study their phenomena.

**Summary Of The Paper:**

This paper introduces a dynamic programming algorithm ("score iteration") which can efficiently compute Q-gradients, and thereafter, "Bellman scores". This Bellman score is useful because it directly gives information about how changes in reward affects policy. The authors then demonstrate its utility for doing behaviour imitation, policy transfer, and counterfactual predictions.

**Summary Of The Review:**

A clearminded paper that would be of interest to the community and should hopefully attract further investigation into the benefits of Bellman score in RL, and even better ways to efficiently compute it.

---

> ### Author Response · Authors · 2022-11-08
> **Author Response**
>
> We thank the reviewer for their time and effort into providing feedback on our paper. We are pleased that the reviewer found our work to be novel, clear minded, and conceptually solid. We will be releasing the source code shortly, which will address the reviewers' advice about providing details on the exact implementation. In light of this, we hope the reviewer would consider raising their score. Please let us know if there are any more points we can address.

---

> > ### Comment · Reviewer_1Fw8 · 2022-11-14
> > **response**
> >
> > My concerns have been addressed. I've increased my score.Good work.

---

### Official Review · Reviewer_gbZN · 2022-11-02

**Confidence:** 4
**Correctness:** 3
**Technical Novelty And Significance:** 4
**Empirical Novelty And Significance:** 2
**Recommendation:** 6

**Clarity, Quality, Novelty And Reproducibility:**

The work has considerable novelty and much potential to impact the RL field. The quality of the analysis (W1, W3) and empirical comparison (W2, W4) could be improved. Moreover, the text could benefit from additional proofreading (W5). Finally, the experiments are not reproducible with the provided resources and information (W4).

**Strength And Weaknesses:**

**Strengths**

1. The novelty and contribution of this work are significant.
2. The reward identification and IRL problems are very relevant and have several downstream implications.
3. The evaluation is performed for different problems, highlighting the generality of the procedure.

**Weaknesses/Areas of improvement**

1. I believe the paper does not appropriately address the limitations of the proposed algorithm in continuous/high-dimensional environments. I would imagine challenges related to the deadly triad [1] would provide even greater instabilities when applied to an estimator of a high-dimensional vector. Moreover, using a backup operator with a buffer of stored behavior will incur the 'double-sampling' issue in stochastic environments (e.g. [1, 2]). Currently, there only is a brief mention of the limitations in the Appendix, solely listing minor and non-specific computation-time considerations.

2. I have some concerns about the empirical IRL results. In particular, the authors mention in the Appendix they perform a hyper-parameter search for their method and run it until convergence. In contrast, the authors mention that the baselines use off-the-shelf implementations that were already tuned for the considered environments. However, since they mainly consider evaluating given unusually low amounts of data, I believe that also the IRL baselines' hyper-parameters should be re-tuned for this specific experimental setting and be allowed to run until convergence (of the reward/agent behavior).

3. Some of the stated contributions sound a bit exaggerated without proper context. For instance, the authors claim "[We] derive, for the first time, the gradient of the Maximum Entropy IRL (Ziebart et al., 2008; Finn et al., 2016b; Fu et al., 2017) objective in the fully general setting with stochastic dynamics case and non-linear reward models." I believe this result (Theorem 2) already follows quite intuitively from the work of Ziebart et al. 2010 [3], which would be important to explicitly specify.

4. The authors state "Hyperparameter ablation studies as well as compute time comparisons are in Appendix C." However, Appendix C does not contain any table/performance curve showing empirical results, but only two paragraphs with some general considerations and recommendations. Such missing information would be very important to get an intuition for the sensitivity and generality of the proposed algorithm. Since the authors state "More detailed ablation results will be added shortly.", do they plan to add this data for the rebuttal revision?

4. I believe this work provides far too little information to allow the reproduction of the results. I find this concern of particular relevance for correctly assessing the contribution, given its significant novelty. As the authors mention that "The implementation code will be released shortly.", I would appreciate it if this code could be anonymously shared with the reviewers for the rebuttal phase.

5. I think there is significant room to improve the paper's writing. For instance, I encountered repetitions of sentences even within the same paragraph, e.g. "The theory in this section will be derived in the Maximum Entropy RL (MaxEntRL) setting (Haarnoja et al., 2017; Kostrikov et al., 2020) but similar results for the vanilla RL setting without entropy regularization can be found in the Appendix [...] The theory in the remaining sections will be derived in the MaxEntRL setting for readability. Similar results for the standard RL setting without entropy regularization can be found in the Appendix." Moreover, I found several typos, e.g.: "i.e gradient of", "Gradients of the objective in Eq. 6 is highly", "the jacobian", "The key challenges with this approach are the difficulty of",... I would encourage the authors to perform additional proofreading of the text to improve clarity.

*References*

[1] Sutton, Richard S., and Andrew G. Barto. Reinforcement learning: An introduction. MIT press, 2018.

[2] Zhu, Yuhua, and Lexing Ying. "Borrowing from the future: An attempt to address double sampling." Mathematical and scientific machine learning. PMLR, 2020.

[3] Ziebart, Brian D. Modeling purposeful adaptive behavior with the principle of maximum causal entropy. Carnegie Mellon University, 2010.

**Additional questions**

1. The underlying assumption of the proposed method is that the observed behavior was generated by a MaxEnt policy. What would the authors empirically expect to observe if this assumption was broken? Would there be any way to verify if this were the case?

2. Given we might be interested in imitating some demonstrations observing an external agent, without access to the agent's actions, can we leverage the same procedure to recover the score of a hypothetical reward function only dependent on the visited states?


**Summary Of The Paper:**

To recover the parameters of an unknown reward function, this work introduces the notion of the Bellman score - corresponding to the (vector-valued) gradients of the log probabilities of the true optimal policy with respect to the true unknown reward. Hence, it proposes an algorithm to estimate this quantity via approximate dynamic programming procedures. The authors show improvements over prior literature performing inverse reinforcement learning and transfer learning in a continuous control benchmark. They also show their algorithm is effective for reward estimation and predicting changes in optimal behavior in an econometrics-based simulation environment.

**Summary Of The Review:**

I found the contribution of this work to be novel and relevant, with the potential to have significant implications. However, in its current form, I believe there are several flaws in the paper, undermining its quality, clarity, and potential fairness. Moreover, given the nature of the contribution, I think that providing a way to reproduce and verify the experiments is of particular importance. Hence, I believe this paper is currently borderline, but I am very open to updating my score based on the rebuttal.

---

> ### Author Response · Authors · 2022-11-15
> **Author Response**
>
> We thank the reviewer for their valuable time and effort into providing feedback on our work. We are pleased that the reviewer found our work to have considerable novelty and much potential impact to the RL field. We hope our response along with the revisions will fully address all of the reviewer's points.
>
> Q. Add a more complete description of the limitations, e.g deadly triad and double sampling issues.
>
> We agree that the deadly triad and double-sampling are limitations worth specifying. We also note, however, that the reward parameter vector should be expected to be relatively low-dimensional for many applications, e.g no more than 64-dimensional in all of our experiments (reward for visual tasks share encoder). This is because true rewards are often parsimonious descriptions of desired behaviors specified by humans, and as a result, are unlikely to require a complex function approximator to estimate. Furthermore, the double-sampling problem is not unique to our work and shared by most modern RL algorithms as they utilize a replay buffer. To address the reviewer’s concern, we will add a more comprehensive description of the limitations of this work, including the deadly triad and double-sampling, to the Limitations/Future works section (currently Conclusions) and the Appendix.
>
> Q. Baseline hyperparameters should be re-tuned for this specific experimental setting
>
> In Appendix B, we had specified that an exhaustive search was conducted to retune the hyper-parameters for the IRL baselines GCL and AIRL. For DAC, we used the default hyper-parameters as we did not find superior parameters upon an exhaustive search. We posit that this is due to the original experiments (https://openreview.net/pdf?id=Xe5MFhFvYGX) already running in the low data regime of 10 trajectories for the same control tasks.
>
> Q. Explicitly specify connection to Ziebart et al. 2010 [3]
>
> The relevant parts from the reviewer’s reference [3] (i.e Ziebart’s thesis) are also in our cited reference (https://www.cs.cmu.edu/~bziebart/publications/maximum-causal-entropy.pdf). We had explicitly stated how our results connect to them in the paragraph below Theorem 2, namely that Eq (8) reduces to the expected feature gap expression. To fully address the reviewer’s concern, we have added a more detailed description of the connection at the end of section A in the Appendix.
>
> Q. More detailed ablation results?
>
> In the revision, we have added complete ablations results to Figure 3 in Appendix C.
>
> Q. Code release?
>
> We will prepare an anonymous repo and update this response with the link shortly. The full code base will be ready to be released in the next couple weeks.
>
> Q. Typos and redundant sentences
>
> Thank you for catching these. We have revised all the typos pointed out by the reviewer, and will continue to proof-read to further improve the clarity of the manuscript.
>
> Q. What happens if the observed behavior is not generated by a MaxEnt policy? How can we verify whether the assumption is broken?
>
> In practice, even the optimal policies trained with a MaxEntRL algorithm (e.g SAC) are not truly MaxEnt optimal and are often indistinguishable from policies trained with non MaxEntRL algorithms. So empirically, GAC works well even if the MaxEnt assumption is broken. For the second question, as long as the demonstrator’s policy probabilities have full support, it is difficult to verify whether the true data generating process involved a MaxEnt optimal policy since any fully supported policy is MaxEnt optimal with respect to the reward $r(x_t, a_t) = \log \pi(a_t | x_t)$
>
> Q. Can we recover state-only rewards when demonstrations don’t contain actions?
>
> Yes. We may compute the first term of Eq. (8) by directly taking state samples $x$ from the demonstrations $D$ and taking their reward gradient, i.e $E_{x \sim D}[\nabla_{\theta} r_{\theta}(x)]$. For the second term of Eq. (8), we may use score-learning to estimate the Q-gradient of the agent $g_{\psi}(x, a) \leftarrow \nabla_{\theta}  r_{\theta}(x) + \bar{g}_{\psi}(x, a)$.

---

> > ### Comment · Reviewer_gbZN · 2022-11-17
> > **Further comments**
> >
> > Thank you for your reply and for acknowledging my feedback. I will address your main responses in order.
> >
> > Response 1) From the supplementary material, I read "The reward r is represented by a single layer network with 64 hidden units, ReLU activations, and a sigmoid output head" To me, this seems to imply that the number of parameters is (input dimension + 1) x 64. Yet, you just claimed that the reward parameter vector is "no more than 64-dimensional" in all experiments. Am I misunderstanding something? Moreover, even if it is 64-dimensional, this is still much considerably higher than 1-dimensional quantities usually estimated via value-functions neural networks. I would expect there to be potentially much more severe stability limitations in complex/stochastic problems. Some formal discussion and analysis about this would be very important to include. Moreover, your main evaluation is performed on deterministic Mujoco environments, hence side-stepping issues related to double-sampling due to stochasticity. I believe these to be important limitations that I cannot find addressed anywhere in the current revision.
> >
> > Response 2) Which exact values/parameter combinations were swept for the different algorithms?
> >  I feel the authors should provide details of the exhaustive search, including tables/graphs in the appendix with all the raw results, even if only from a limited number of seeds. This would be very helpful for the IRL community.
> >
> > Response 3) My concern was with the form in which some of the results were presented, which I believe possibly overclaims their contribution, given their strong connection to existing prior analysis (see review). Unfortunately, I do not find that the used language has changed in the current revision.
> >
> > Response 4) The added ablations appear to consider a very minimal subset of the hyper-parameters and only report aggregate statistics. It's hard to draw conclusions without more comprehensive, per-environment results. Moreover, again no precise information is given about compute time comparison.
> >
> > Response 5) I hope the authors will shortly share this with the reviewers, as the current version of the submission is still not reproducible.
> >
> > Overall, I do not feel my concerns were addressed. While I appreciate the idea and contribution, I still believe the paper has a very large room for improvement. I am not convinced that the current version is ready to be accepted.

---

> > > ### Author Response · Authors · 2022-11-19
> > > **Author Response**
> > >
> > > We thank the reviewer for their enthusiastic engagement to improve the manuscript. We have addressed their latest concerns in order. In light of this, we hope the reviewer would consider raising their score. We are happy to incorporate additional requests for revision after the rebuttal period if the reviewer has any more suggestions.
> > >
> > > Response 1) The reviewer is correct about the number of reward parameters. We have added a discussion of the deadly triad and double-sampling issues to the limitations section in Appendix D. We are happy to incorporate the reviewer’s feedback on this section and incorporate them into the camera-ready if the manuscript is accepted.
> > >
> > > Response 2) We will have to rerun the hyperparameter sweep for the baselines in order to produce detailed tables/graphs of the results. We will have them ready for the camera-ready version if the manuscript is accepted.
> > >
> > > Response 3) In the latest revision, we have changed the language in the direction that the reviewer has suggested.
> > >
> > > Response 4) We have added per-environment ablation results on the visual control tasks to Appendix C. More detailed compute time comparison results have been added to Appendix D. We will continue to update the ablation results with more hyper-parameter values and environments as they become available.
> > >
> > > Response 5) We will shortly update https://github.com/reviewanon/gac for the review period. The full code will be released in the next couple weeks.

---

> > > > ### Comment · Reviewer_gbZN · 2022-11-24
> > > > **Further comments**
> > > >
> > > > In the latest revision, the added discussion about the deadly triad and double-sampling issues is severely limited and uninformative, only briefly mentioning them in the Appendix. I believe some formal discussion and analysis about these concerns would be very important to include in the main text. Analogously, I did not find most of my other concerns to have been properly addressed. As of this comment's date, the provided code link still points to an empty repository. It would have been important to provide a version of the code early in the review process to allow for the correct assessment of the implementation's correctness and the results reproducibility.
> > > >
> > > > For these reasons, I am still not convinced this work is ready to be published and will not raise my score.

---

> > > > > ### Author Response · Authors · 2022-12-11
> > > > > **Author response continued**
> > > > >
> > > > > We thank the reviewer for their additional feedback. We apologize for the delay in response. We believe there were issues with the open review email notification system as the first author was not receiving notifications about reviewer comments posted after the rebuttal period.
> > > > >
> > > > > We are happy to include a more formal discussion and analysis about the deadly triad and double-sampling issues in the main text of the camera-ready if the paper is accepted. We can provide analyses in a similar fashion to those in https://arxiv.org/abs/1812.02648. If the reviewer has specific suggestions for what analyses they would like to see, please do not hesitate to let us know. Regarding the broken code link, we will fix it ASAP within the next 24 hours and follow-up.

---

> > > > > ### Author Response · Authors · 2022-12-12
> > > > > **Code link fixed**
> > > > >
> > > > > We thank the reviewer for their patience. We have created a new repo to fix the broken code link. Please find the code here: https://github.com/reviewanon/gac_temp

---

> > > > > ### Author Response · Authors · 2022-12-14
> > > > > **One more fix to the code**
> > > > >
> > > > > We discovered that in the codebase push made two days ago, we had mis-specified the reward learning rate in the hyper-parameter file which will not generate the correct results. We have just pushed the changes with the corrected hyper-parameters in the config file.

---

### Decision · Program_Chairs · 2023-01-20

**Decision:**

Accept: notable-top-25%

**Justification For Why Not Higher Score:**

The paper could possibly be given a score of "Accept (oral)" based on score calibration with other accepted papers.

**Justification For Why Not Lower Score:**

"Accept (spotlight)" is justified as the reviewers agreed that the paper has high novelty and the results would be of broad interest to the community.

**Metareview: Summary, Strengths And Weaknesses:**

The reviewers agreed that the paper proposes a novel gradient-based algorithm for learning an unknown reward function using new insights about the underlying gradient structure. However, the reviewers also raised several concerns and questions in their initial reviews. We want to thank the authors for their responses and active engagement during the discussion phase. The reviewers appreciated the responses, which helped in answering their key questions. As a result, the reviewers have an overall positive assessment, and there is a consensus for acceptance. The reviewers have provided detailed feedback, and we strongly encourage the authors to incorporate this feedback when preparing the final version of the paper.

**Note From Pc:**

if the above contains the word "oral" or "spotlight" please see: "oral" presentation means -> notable-top-5% and "spotlight" means -> notable-top-25%. As stated in our emails, we are disassociating presentation type from AC recommendations